# Visual Concepts Tokenization

**Tao Yang**[1*]**, Yuwang Wang**[2†]**, Yan Lu**[2]**, Nanning Zheng**[1]
yt14212@stu.xjtu.edu.cn,
{yuwwan,yanlu}@microsoft.com,
nnzheng@mail.xjtu.edu.cn
[1]Xi'an Jiaotong University, [2]Microsoft Research Asia
https://github.com/thomasmry/VCT

## Abstract

Obtaining the human-like perception ability of abstracting visual concepts from concrete pixels has always been a fundamental and important target in machine learning research fields such as disentangled representation learning and scene decomposition. Towards this goal, we propose an unsupervised transformer-based Visual Concepts Tokenization framework, dubbed VCT, to perceive an image into a set of disentangled visual concept tokens, with each concept token responding to one type of independent visual concept. Particularly, to obtain these concept tokens, we only use cross-attention to extract visual information from the image tokens layer by layer without self-attention between concept tokens, preventing information leakage across concept tokens. We further propose a Concept Disentangling Loss to facilitate that different concept tokens represent independent visual concepts. The cross-attention and disentangling loss play the role of induction and mutual exclusion for the concept tokens, respectively. Extensive experiments on several popular datasets verify the effectiveness of VCT on the tasks of disentangled representation learning and scene decomposition. VCT achieves the state of the art results by a large margin.

## 1 Introduction

Despite the remarkable success of deep learning in various vision tasks, such as classification [21, 11], detection [35, 7], segmentation [32, 40], it still suffers from the requirement of a tremendous amount of training data [44], low robustness and generalization [43, 19], and lack of interpretability [1]. Those traditional vision tasks learn the visual concepts, such as semantics and object localization, from artificially predefined guidance. On the contrary, human is capable of extracting abstract concepts from concrete visual signals, then using those visual concepts to understand and depict the world comprehensively. Towards learning visual concepts from observations, Bengio et al. [4] propose disentangled representation learning to discover the visual concepts as the explanatory factors hidden in the observed data. To achieve disentangled representation, the following works, based on VAE [22, 5, 28, 27, 8] or GAN [9], rely on probabilistic-based regularization of the latent space. However, it has been fundamentally proved to be impossible if only relying on probability constrains [30]. Introducing inductive bias is necessary to solve the identifiability issue. Besides, those methods often fail in complex scenes with multiple objects. To address those complex scenes, another branch of learning visual concepts aims to spatially decompose a scene image into different object regions represented as different segmentation masks [18, 6]. Typically, these methods rely on explicit spatial decomposition and can not learn global visual concepts. In this paper, we are particularly interested

---

*Work done during internships at Microsoft Research Asia.

†Corresponding author

36th Conference on Neural Information Processing Systems (NeurIPS 2022).

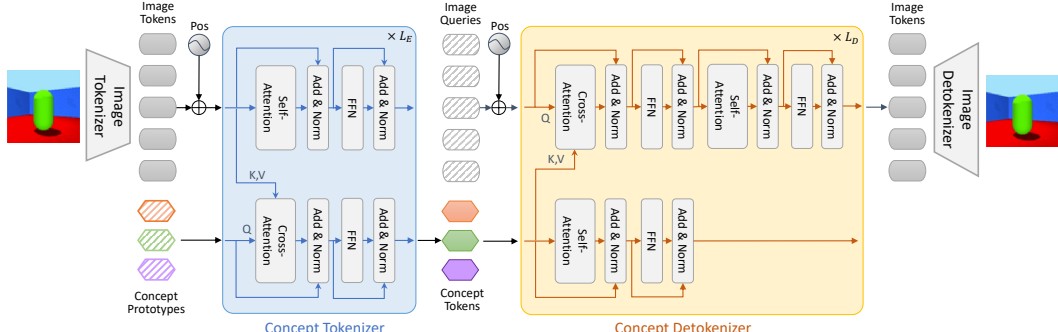

Figure 1: The framework of Visual Concept Tokenization (VCT). An image is represented as a set of concept tokens, and each token reflects a visual concept, such as green object color, blue background color. The concept prototypes and image queries are shared across different images.

in finding a general way to learn visual concepts from pixels, which covers the aforementioned two branches, disentangled representation learning and scene decomposition.

Originated from Natural Language Processing (NLP), tokenization means the process of demarcating a string of input characters into sections named tokens. For the visual signal, we propose an unsupervised transformer-based approach, Visual Concept Tokenization (VCT), to extract the visual concept inside a given image as a set of tokens, serving as a general solution for visual concept learning. For example, given an image of a 3D scene [27], our VCT can represent it into the object shape token, object color token, object scale token, background color token, floor color token and pose token. Before introducing the details of VCT, we would like first to discuss the requirements of the tokens to achieve a good representation of the visual concepts contained in an image. Those tokens should satisfy the following three conditions: $(i)$ *completeness*, one can reconstruct the image with those tokens; $(ii)$ *disentanglement*, different tokens should represent independent visual concepts, and each token should only reflect one kind of visual concept variation; $(iii)$ *disorder*, considering the disordered nature of concepts, the ranking order of tokens should not carry any information. We refer to the tokens as concept tokens if they satisfy the above three conditions. Note that the concept tokens are different from image tokens, i.e., the input of vision transformers, which are simply the grid feature obtained by a CNN backbone [23] or patch features from a linear projection layer [11]. Those image tokens are typically entangled in terms of abstract information and positionally sensitive.

To obtain concept tokens from images, we build a novel transformer-based architecture, as shown in Figure 1, consisting of a Concept Tokenizer and a Concept Detokenizer. The Concept Tokenizer abstracts visual concepts from image tokens, and the Concept Detokenizer reconstructs image tokens to meet the *completeness* condition. In the Concept Tokenizer, using learnable concept prototypes as query, we use cross-attention to induct information from image tokens layer by layer independently, without any interference between tokens of concept part, to meet the *disentanglement* requirement. It is worth noticing that, although the concept prototypes are dataset-level shared, different from other works containing learnable dataset-level queries with interaction (self-attention), such as DETR [7], Perceiver [24], Retriever [42], there is no self-attention across our concept queries or tokens. Finally, considering disorder, since there is no interference between the concept tokens, the order of the tokens carries no information in the tokenization process. In addition, no position embedding is added to concept tokens to maintain this disorder nature for the Concept Detokenizer.

To further facilitate the *disentanglement* of concept tokens, inspired by DisCo [36] which achieves disentanglement via contrasting the visual variations produced by a pretrained generative model, we propose a Concept Disentangling Loss to encourage the mutual exclusivity between the visual variations caused by modifying different concept tokens. Specifically, we modify a concept token and result in image variation. After feeding to the Concept Tokenizer, by minimizing the Concept Disentangling Loss, the image variation should only affect the modified concept token.

We verify the effectiveness of VCT on disentanglement and scene decomposition tasks, and both achieve state-of-the-art performance with a significant improvement. Surprisingly, we find that visual concept tokens are well aligned with language representations by simply adopting the CLIP [33] image encoder as the image tokenizer. Therefore, we can control the image content using text input.

Our main contributions can be summarized as:

- We present a general solution to extract visual concepts from concrete pixels, which can achieve disentangled representation learning and scene decomposition.
- We build an unsupervised framework, including Concept Tokenizer and Detokenizer, to represent an image into a set of tokens, and each token reflects a visual concept.
- We propose a Concept Disentangling Loss to facilitate the mutual exclusivity of the visual concept tokens.

## 2 Related Works

**Image Tokenization**  For visual inputs, the previous methods to get image tokens can be divided into region-based, grid-based and patch-based. For region-based, each token responds to an object region predicted by a detector [2]. For grid-based, each token is a spatially $1 \times 1$ vector taking from the extracted feature of a CNN backbone [23]. For the patch-based, the image is first divided into patches, and each token corresponds to the feature extracted from one patch [11]. Our visual concept tokenization is different from the previous works due to it is the process of extracting high-level visual concepts from the image tokens produced by previous image tokenization methods. We can adopt the grid-based or patch-based image tokenization as the module before concept tokenization. The region-based one needs an extra detector to get object-level semantics, which is redundant and not desired in our framework. We refer to the reverse process of image tokenization and concept tokenization as image detokenization and concept detokenization, respectively.

**Disentangled Representation Learning**  Disentangled representation learning is introduced in Bengio et al. [4], which aims to discover the hidden explanatory factors hidden in the observed data. Each dimension of the disentangled representation corresponds to one independent factor. Following that, there are some VAE-based works that achieve disentanglement [22, 5, 28, 27, 8] by relying on probability-based regularizations on the latent space. Locatello et al. [30] prove that only these regularizations are not enough for disentanglement, and extra conductive bias on model and data is required. Following works explore an alternative way for disentangled representation, including leveraging pretrained generative model [36] or symmetry properties modeled with group theory [41]. To the best of our knowledge, our VCT is the first Transformer based framework, which takes advantage of the cross attention mechanism to discover the factor as well as forbid the entanglement across the concept tokens. This specific architecture serves as network inductive bias. VCT significantly improves the disentanglement ability, especially on some challenging datasets.

**Scene Decomposition**  Scene decomposition, also referred to as object-centric representation, aims to decompose a scene into objects, each with spatial segmentation masks [6]. Different from previous disentangled representation learning methods, which typically only consider one main object, the visual concepts on scene decomposition tasks focus on the object level representation. One can achieve scene segmentation with explicit pixel-level supervision [32, 40]. To get rid of the supervision, MONet [6] proposes an attention network to infer the mask and representation of each object from an image in an autoregressive way. Locatello et al. [31] propose a slot attention module to insert the image feature, which is extracted by a CNN backbone, into slots. This process is similar but fundamentally different from the cross-attention [3] we used for concept tokens. The key is that slot attention applies a softmax normalization in the slot dimension. The slots compete and interact with each other when including the image feature. The previous works need an explicit pixel-level mask specifically designed for the decomposition task. However, our concept token representation is a general and abstract high-level visual representation, and we would like to show that without any dedicated design, VCT can achieve scene decomposition and represent each object as a concept token. We also provide a solution to get a pixel-level mask in the experiment. Some other works explore the loss functions, such as energy-based loss [12] and contrastive loss [3]. Our Concept Disentangling Loss is based on the manipulating concept tokens, and is specifically designed for VCT.

## 3 Visual Concept Tokenization

Given an image set $\{x_i\}$, our target is to represent each image $x_i$ as a set of $M$ concept tokens of $D$ dimension $C_i \in \mathbb{R}^{M \times D}$, where each row of $C_i$ represents a concept token, and each token can

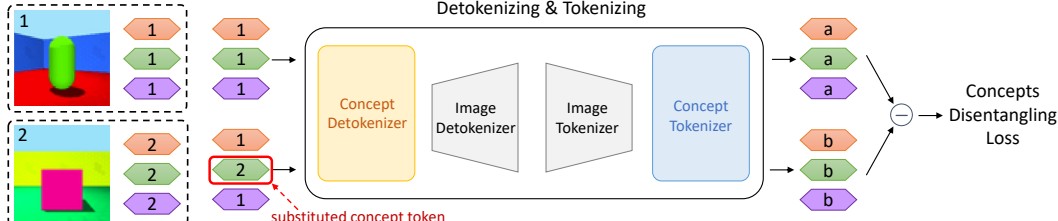

Figure 2: Illustration of Concept Disentangling Loss. The second concept token (labeled in green color) is substituted (from 1 to 2) to create the visual variation. Concept tokens {a,a,a} and {b,b,b} are the outputs for inputing {1,1,1} and {1,2,1} to detokenizing and tokenizing, respectively.

reflect only one visual concept contained in the image set, such as object color red, background color green, object shape cube. We first present an overview of VCT, then introduce the Concept Tokenizer, Concept Detokenizer, and Concept Disentangling Loss sequentially.

## 3.1 Overview of VCT

The overview framework of VCT is shown in Figure 1. An image $x_i$ is first be tokenized into a number of $N$ image tokens, $Z_i \in \mathbb{R}^{N \times D}$, using an image tokenizer $\mathcal{I}_T$, where each row of $Z_i$ is an image token. The image tokenizer $\mathcal{I}_T$ could be a pre-trained or randomly initialized module, such as a pretrained VQ-VAE [38] encoder, CLIP [33] image encoder, CNN-based encoder. Then each image token is added with a positional embedding to include 2D spatial information in the same way as [24]. Then a Concept Tokenizer $\mathcal{V}_T$, as a key component of VCT, takes the image tokens $Z_i$ as input, as well as a set of $M$ concept prototypes $P \in \mathbb{R}^{M \times D}$. $P$ is learnable and dataset-shared, and each row of $P$ is a prototype of concept attribute, such as object color, background color, object shape. With prototype $P$, Concept Tokenizer $\mathcal{V}_T$ extracts the visual concepts from image tokens $Z_i$, resulting in the set of concept tokens $C_i = \mathcal{V}_T(P, Z_i)$. For example, the object in two images $x_i$ and $x_j$ are red and green. $p^1$ (first row of $P$) is the color prototype. The corresponding extracted concept tokens $c_i^1$ (first row of $C_i$) and $c_j^1$ (first row of $C_j$) represent the red and green colors, respectively. Then the concept tokens $C_i$ are fed into a Concept Detokenizer $\mathcal{V}_D$. The Concept Detokenizer $\mathcal{V}_D$ reconstructs the image tokens $Z_i$, with an extra input of dataset-shared learnable image queries $Y \in \mathbb{R}^{N \times D}$. The reconstructed image tokens are decoded into pixels via an image detokenizer $\mathcal{I}_D$, which is an inversing module of image tokenizer $\mathcal{I}_T$. For example, one can choose a pre-trained VQ-VAE encoder and decoder for the image tokenizer and detokenizer, respectively.

## 3.2 Concept Tokenizer

Given an image $x_i$, the Concept Tokenizer $\mathcal{V}_T$ extracts the concept tokens $C_i$ from the image tokens $Z_i$ and prototypes $P$. The detailed implementation of Concept Tokenizer $\mathcal{V}_T$ is shown in Figure 1. To satisfy the aforementioned requirements for visual concept representation in Section 1, we adopt two different types of attention for the image tokens and concept prototype tokens, respectively. Specifically, for the image tokens, we adopt a standard Transformer layer using self-attention to process image tokens, followed by a feed-forward network (FFN) layer. For the concept part, to induct information from image tokens and prevent interference between concept tokens, we adopt cross-attention$(Q, K, V)$ without following self-attention. In the cross-attention operation, $Q$ is the tokens from the concept part, $K$ and $V$ are the tokens from the image part. For the first layer, the cross-attention block takes concept prototypes $P$ as $Q$, and image tokens $Z_i$ as $K$ and $V$, resulting in cross-attention$(P, Z_i, Z_i)$. Each cross-attention block is followed by an FFN. To provide pathways for local and global visual concepts flexibly, we use a stack of $L_E$ layers of the aforementioned self-attention & FFN, cross-attention & FFN layers to extract visual concepts from image tokens layer by layer. It is worth noting that the tokens in the concept part are only used as queries, and there is no interaction between those queries. Consequentially, the encoding process of each token is independent: $c_i^j = \mathcal{V}_T(P, Z_i)^j = \mathcal{V}_T(p^j, Z_i)$. Note that $\mathcal{V}_T(\pi(P), Z_i) = \pi(\mathcal{V}_T(P, Z_i))$, where $\pi$ is shuffle function, disorder nature is satisfied for Concept Tokenizer. This is the key to ensuring disentanglement, serving as a network inductive bias.

### 3.3 Concept Detokenizer

Given the concept tokens $C_i$, the Concept Detokenizer $\mathcal{V}_D$ reconstructs the image tokens $Z_i$. The detailed implementation of Concept Detokenizer is shown in Figure 1. As a reverse procedure of Concept Tokenizer, for the Concept Detokenizer we follow a symmetry design of the Concept Tokenizer. Similar to the concept prototypes $P$ used to query concept tokens in the Concept Tokenizer, we adopt an array of $N$ image queries $Y \in \mathbb{R}^{N \times D}$ to query visual information contained in the concept tokens $C_i$. Those image queries $Y$ act as placeholders of features shared across the dataset. We insert the image-specific visual information carried by the concept tokens into $Y$ via cross-attention$(Q, K, V)$ by using image queries $Y$ as $Q$, and concept tokens $C_i$ as $K$ and $V$, resulting in cross-attention$(Y, C_i, C_i)$. Different from the disentanglement requirement in the Concept Tokenizer, in Concept Detokenizer, we need to mix up the isolated visual information in different concept tokens to reconstruct the image tokens. Thus, we adopt self-attention after the aforementioned cross-attention$(Y, C_i, C_i)$, as well as to fuse the concept tokens. There is an FFN after each attention operation. We stack $L_D$ layers of the aforementioned structure to reconstruct image tokens.

### 3.4 Concept Disentangling Loss

The Concept Tokenizer structure can ensure there is no interference between the concept tokens. We use the Concept Disentangling Loss to encourage the mutual exclusivity of concept tokens. Generally, the Concept Disentangling Loss is a cross-entropy loss on the image variation caused by manipulations of the concept tokens. We demonstrate the process of calculating Concept Disentangling Loss in Figure 2. The two steps are: $(i)$ producing image variations by substituting one concept token, $(ii)$ identifying the image variations. Firstly, we introduce the process of producing image variations. We introduce image variation by replacing specific concept token. Specifically, we introduce the detailed implementation given a batch of images $\{x_i\}^B$. For image $x_i$, we randomly select another image $x_j, j \neq i$. The related two sets of concept tokens are $C_i$ and $C_j$, respectively. We randomly select an index $l, 0 \leq l < K$, and replace $c_i^l$ (the $l$-th token of $C_i$) with $c_j^l$, resulting in $\hat{C}_i$. Then we decode $C_i$ and $\hat{C}_i$ to images via Concept Detokenizer $\mathcal{V}_D$ and image detokenizer $\mathcal{I}_D$ sequentially. The image variation between the reconstructed image pair $x_i' = \mathcal{I}_D(\mathcal{V}_D(C_i))$ and $\hat{x}_i' = \mathcal{I}_D(\mathcal{V}_D(\hat{C}_i))$ is caused by the aforementioned replacing operation $(c_i^l \rightarrow c_j^l)$. The next step is to identify which concept token is replaced. We first encode the images $x_i'$ and $\hat{x}_i'$ to get concept tokens via image tokenizer $\mathcal{I}_T$ and Concept Tokenizer $\mathcal{V}_T$. Then the concept token variation is

$$\Delta C = \mathcal{V}_T(\mathcal{I}_T(x_i')) - \mathcal{V}_T(\mathcal{I}_T(\hat{x}_i')). \tag{1}$$

The Concept Disentangling Loss is

$$\mathcal{L}_{dis} = CrossEntropy(norm(\Delta C), l), \tag{2}$$

where $norm(\Delta C)$ means to calculate the $\ell_2$ norm for each row, i.e., the norm of the variation of each token, resulting in a vector of $K$ dimension. $l$ is the ground truth. It is a one-hot vector of $K$ dimension, with the replaced dimension set to 1. To prevent disentangling loss from sacrificing the reconstruction quality, we only optimize disentangling loss w.r.t. Concept Tokenizer.

### 3.5 Total Loss

Besides the aforementioned Concept Disentangling Loss, we also need the reconstruction loss $\mathcal{L}_{rec}$ to reconstruct the image tokens. We can choose the formulation of $\mathcal{L}_{rec}$ according to the choice of image tokenizer and detokenizer. For example, when using VQ-VAE, we can predict the quantized label of each reconstructed image token and use cross-entropy between the predicted label and ground truth label as the reconstruction loss. For other image tokenizer and detokenizer, we adopt MSE between the image and reconstructed image as reconstruction loss. Together with the Concept Disentangling Loss $\mathcal{L}_{dis}$, the total loss is $\mathcal{L} = \mathcal{L}_{rec} + \lambda_{dis}\mathcal{L}_{dis}$, where $\lambda_{dis}$ is the hyper-parameter. We set $\lambda_{dis} = 1$ and adopt VQ-VAE for $\mathcal{L}_{rec}$ in all the experiments.

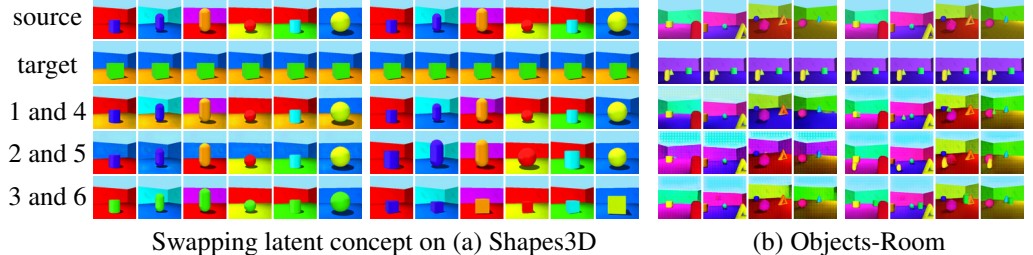

| source | | | |
| target | | | |
| 1 and 4 | | | |
| 2 and 5 | | | |
| 3 and 6 | | | |

Swapping latent concept on (a) Shapes3D    (b) Objects-Room

Figure 3: Visualization of swapping concepts on Shapes3D and Objects Room. The images in first row provide source concepts, and the second provides target concept. Rest of images are swapped ones ("1 and 4" represent that the row is corresponding to concept 1 swapped image (left) and concept 4 swapped image (right)). More results are provided in Appendix B.

Table 1: Comparisons of disentanglement on the FactorVAE score and DCI disentanglement metrics (mean $\pm$ std, higher is better). VCT achieves the state of the art performance with a large margin in almost all the cases compared to all of the baselines. Especially on the MPI3D dataset. For MIG and BetaVAE metrics and other details of experiments, please see Appendix A.

| Method | Cars3D | | Shapes3D | | MPI3D | |
| --- | --- | --- | --- | --- | --- | --- |
| | FactorVAE score | DCI | FactorVAE score | DCI | FactorVAE score | DCI |
| *VAE-based:* | | | | | | |
| FactorVAE | $0.906 \pm 0.052$ | $0.161 \pm 0.019$ | $0.840 \pm 0.066$ | $0.611 \pm 0.082$ | $0.152 \pm 0.025$ | $0.240 \pm 0.051$ |
| $\beta$-TCVAE | $0.855 \pm 0.082$ | $0.140 \pm 0.019$ | $0.873 \pm 0.074$ | $0.613 \pm 0.114$ | $0.179 \pm 0.017$ | $0.237 \pm 0.056$ |
| *GAN-based:* | | | | | | |
| InfoGAN-CR | $0.411 \pm 0.013$ | $0.020 \pm 0.011$ | $0.587 \pm 0.058$ | $0.478 \pm 0.055$ | $0.439 \pm 0.061$ | $0.241 \pm 0.075$ |
| *Pre-trained GAN-based:* | | | | | | |
| LD | $0.852 \pm 0.039$ | $0.216 \pm 0.072$ | $0.805 \pm 0.064$ | $0.380 \pm 0.062$ | $0.391 \pm 0.039$ | $0.196 \pm 0.038$ |
| CF | $0.873 \pm 0.036$ | $0.243 \pm 0.048$ | $0.951 \pm 0.021$ | $0.525 \pm 0.078$ | $0.523 \pm 0.056$ | $0.318 \pm 0.014$ |
| GS | $0.932 \pm 0.018$ | $0.209 \pm 0.031$ | $0.788 \pm 0.091$ | $0.284 \pm 0.034$ | $0.465 \pm 0.036$ | $0.229 \pm 0.042$ |
| DS | $0.871 \pm 0.047$ | $0.222 \pm 0.044$ | $0.929 \pm 0.065$ | $0.513 \pm 0.075$ | $0.502 \pm 0.042$ | $0.248 \pm 0.038$ |
| DisCo | $0.855 \pm 0.074$ | $0.271 \pm 0.037$ | $0.877 \pm 0.031$ | $0.708 \pm 0.048$ | $0.371 \pm 0.030$ | $0.292 \pm 0.024$ |
| *Concept-based:* | | | | | | |
| COMET | $0.339 \pm 0.008$ | $0.024 \pm 0.026$ | $0.168 \pm 0.005$ | $0.002 \pm 0.000$ | $0.145 \pm 0.024$ | $0.005 \pm 0.001$ |
| VCT (Ours) | $\mathbf{0.966 \pm 0.029}$ | $\mathbf{0.382 \pm 0.080}$ | $\mathbf{0.957 \pm 0.043}$ | $\mathbf{0.884 \pm 0.013}$ | $\mathbf{0.689 \pm 0.035}$ | $\mathbf{0.475 \pm 0.005}$ |

## 4 Experiments

### 4.1 Disentanglement Results

In order to verify the disentanglement of the learned visual concepts of our framework, we conduct experiments on the task of disentangled representation learning.

**Datasets** Following [36], we conduct the experiments on the public datasets below, which are popular in disentangled representation literature: **Shapes3D** [27] is a dataset of 3D shapes generated from 6 factors of variation. **MPI3D** [17] is a 3D dataset recorded in a controlled environment, defined by 7 factors of variation, and **Cars3D** [34] is a dataset of CAD models generated by color renderings from 3 factors of variation. We follow the literature to resize images to 64x64 resolution.

**Baselines & Metrics** The baselines contain four different types: The VAE-based baselines are **FactorVAE** [27], and $\beta$-**TCVAE** [8]. The GAN-based baseline is **InfoGAN-CR** [29]. For pre-trained GAN-based baselines, we adopt **GANspace (GS)** [20], **LatentDiscovery (LD)** [39], **ClosedForm (CF)** [37], **DeepSpectral (DS)** [26] and **DisCo** [36]. For the concept-based method, we use **Energy Concepts (COMET)** [13] as our baseline. We follow [30] to conduct our experiments with different random seeds. We have 25 runs for each method. Four popular and representative metrics are used in our experiments: FactorVAE score [27], the DCI [14], $\beta$-VAE score [22], and MIG [8]. However, since concept-based disentangled representations are vector-wise, we follow [13] to perform PCA as post-processing on the representation and evaluate the performance with these metrics.

**Quantitative Results** Table 1 shows the comparison between VCT and other SOTA methods under different disentanglement metrics. VCT achieves superior performance with a large margin, which demonstrates the disentanglement ability of our framework. The baseline methods include four categories. The VAE-based and GAN-based methods suffer from the traded-off between generation and disentanglement [36]. The pre-trained GAN-based methods are limited by the latent space of GAN. COMET adopts energy-based modeling, which is more suitable for composition variations, e.g., scene decomposition. However, our method does not have these limitations. In addition, since our disentanglement is conducted in a learned space, which reduces the difficulty of the problem.

**Qualitative Results** Different from dimension-wise disentanglement methods, concept token is vector-wise disentanglement. Therefore, we can not do the latent traversals as the dimension-wise disentanglement methods did. We swap the concept tokens of different images instead. As Figure 3 shows, VCT is capable of learning pure factors. Note that our framework achieves not only pure disentanglement but also has high-quality reconstruction. For results on real-world datasets CelebA/MSCOCO/KITTI, please see Appendix B. Besides, VCT can combine with pretrained GANs for image editing (details are presented in Appendix B).

## 4.2 Ablation Study

We conduct the ablation study on disentangled representation learning of Shapes3D from the following five aspects. For ease of conducting experiments, we ensure 15 runs for each setting.

**Image Tokenizer Analysis** Since VCT conducts disentanglement on the latent space of an autoencoder, the choice of initialization of encoder influence the performance of the framework. We study the following choices of encoder: randomly initialized naive autoencoder (AE), pretrained autoencoder (pre-trained AE), and pretrained VQ-VAE. The usage of pre-trained VQ-VAE reduced the difficulty of the problem. Table 2 presents the results of different types of image tokenizers. The results show that the model with pre-trained VQ performs best as expected. In addition, VCT still works using a randomly initialized autoencoder, which demonstrates that VCT does not rely on the pre-trained models. Besides, VCT can still disentangle the latent space that is highly entangled (a pre-trained naive autoencoder). Even with the Patch tokenizer in ViT [10], VCT still works.

**Concept Disentangling Loss** We study the effectiveness of Concept Disentangling Loss by removing it, denoted as "wo $\mathcal{L}_{dis}$." The results

Table 2: Ablation study of VCT on image tokenizer, components, batchsize and token numbers.

| Method | MIG | DCI |
| --- | --- | --- |
| Patch + VCT | 0.361 | 0.668 |
| AE + VCT | 0.484 | 0.802 |
| pretrained AE + VCT | **0.560** | 0.849 |
| pretrained VQ-VAE + VCT | 0.525 | **0.884** |
| AE + VCT wo $\mathcal{L}_{dis}$ | 0.165 | 0.692 |
| pretrained VQVAE + VCT wo $\mathcal{L}_{dis}$ | 0.286 | 0.731 |
| w/ self-attention | 0.000 | 0.008 |
| wo detach | 0.392 | 0.871 |
| w/ pos embedding | 0.525 | 0.884 |
| CNN DeTokenizer | 0.157 | 0.847 |
| Transformer DeTokenizer | 0.467 | 0.821 |
| Concept DeTokenizer | **0.525** | **0.884** |
| batchsize = 16 | 0.497 | 0.862 |
| batchsize = 32 | 0.525 | 0.884 |
| batchsize = 64 | **0.535** | **0.900** |
| tokens number = 10 | **0.533** | 0.867 |
| tokens number = 20 | 0.525 | 0.884 |
| tokens number = 30 | 0.493 | **0.885** |

in Table 2 demonstrate that even without Concept Disentangling Loss, our framework still can learn a disentangled representation to a considerable extent, no matter when AE or VQ-VAE is used.

**Concept Tokenizer** As we stated in Section 3, the interference between concepts is vital for VCT. In order to verify it, we add a self-attention after each cross-attention block in Concept Tokenizer, which is denoted as "w/ self-attention." Table 2 demonstrate that the model catastrophic fails. In addition, to verify that only optimizing Concept Tokenizer w.r.t $\mathcal{L}_{dis}$ is effective, we train VCT without stop gradient when computing $\mathcal{L}_{dis}$ and the performance significant drops, which is denoted as "wo detach." Finally, since Concept Tokenizer already satisfied the disorder requirement, adding position embedding on our framework ("w/ pos embedding") only has little influence on the performance.

**Concept Detokenizer** In order to verify the effectiveness of the Concept Detokenizer design, we use a CNN and transformer (self-attention) to replace our Concept Detokenizer, we find that the performance drops significantly, especially on the CNN decoder. We posit the reason is that the non-symmetric architecture and directly decoding from concept tokens make VCT less disentangled.

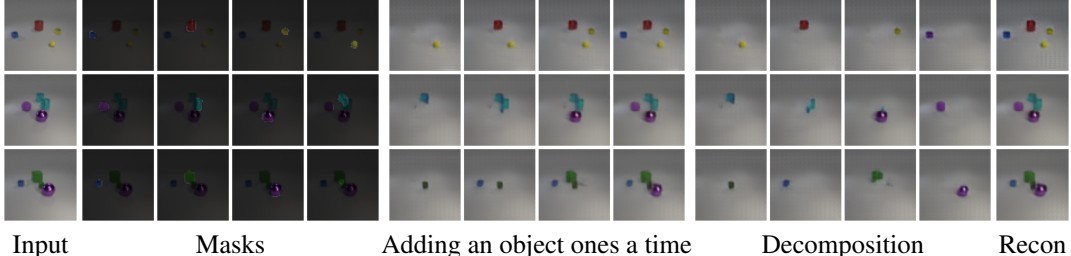

| Input | Masks | Adding an object ones a time | Decomposition | Recon |

Figure 4: Scene decomposition results on CLEVR. By replacing the Concept Token of an image with the token of the object from another image and decoding the tokens, we can add it to the image.

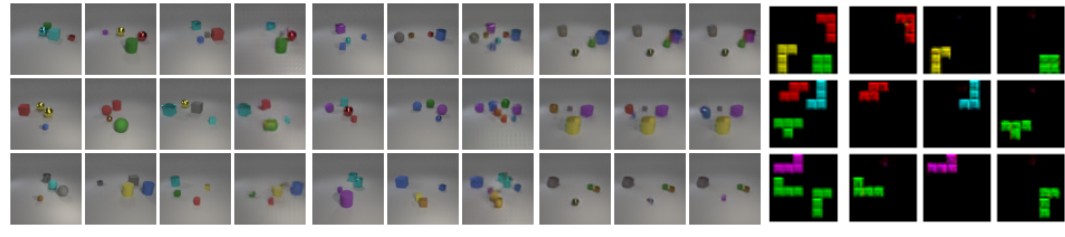

(a1) Light+objects+objects  (a2) Objects+objects    (b) Interpretation       (c) Decomposition

Figure 5: (a) Recombination of objects and background. By replacing the concept tokens of objects, we can add objects from one scene to another. (b) linear interpolation of concept token of an object. (c) decomposition on Teris.

**Sensitive Analysis** The batch size and the number of concepts affect the sample diversity of computing $\mathcal{L}_{dis}$. We also explore of training with different batch sizes and different numbers of concepts. As shown in Table 2, we see that the batch size and concept number slightly influence VCT on the condition that they are larger than the number of ground truth factors (see details in appendix B). Note that VCT works well with a small batch size (32), and the performance only slightly drops even with a batch size of 16. Since we set $\lambda_{dis} = 1$ for all experiments, VCT is robust to the hyper-parameters.

## 4.3   Scene Decomposition

Next, in this section, we verify the ability of VCT to decompose a scene into object-level representations. Note that our requirements of concept tokens are applicable to both object-level and factor-level representations. Therefore, which kind of concept VCT learns is determined by data. Different from COMET [13], there is no need to change the framework to bias the model to learn objects. We evaluate the decomposition ability of our framework on **CLEVR**[25] and **Teris**[18] dataset. Finally, we verify the assumption that the type of concept is driven by the data on the **Objects-Room**[16] dataset, which has both objects and global factors. The datasets used here are all public.

**Decomposition** Given a scene image, VCT represents a single object inside the scene with a single concept token and thus spatially decomposes the scene image into objects. See the details in Appendix C. Figure 4 and 5 (c) illustrate that an individual object is encoded in a single concept token both on **CLEVR** and **Tetris**.

**Quantitative Evaluation** Note that we can not obtain masks from VCT directly. In order to have a quantitative comparison, we decode a explicit mask as did in [31, 18] (see Appendix C for detail). We present the qualitative results of masks in Figure 4. Our framework achieves an ARI [18] of 0.923 and a mean segmentation covering [15](MSC) of 0.760. COMET [13], MONET [6], and Slot Attention [31] obtain ARI of 0.916, 0.873, and 0.818, and obtain MSC of 0.713, 0.701, and 0.750.

**Objects Recombination** As demonstrated in Figure 5, VCT can add an object from one scene to another, remove an object from a scene, and combine different objects from two scenes by simply replacing tokens. We can even generate an image with more objects in one scene than the dataset images, which is out-of-distribution. Please see the details in Appendix C. Compared to COMET and

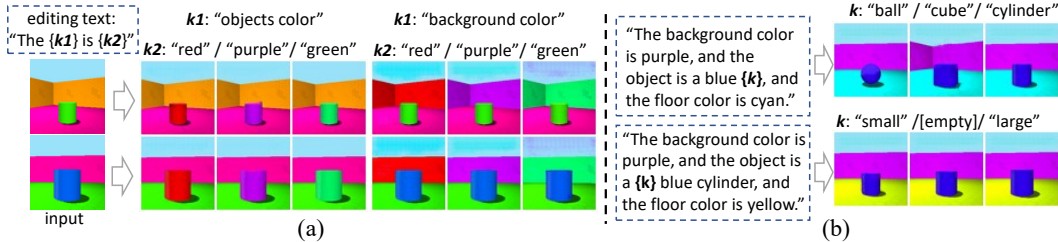

Figure 6: CLIP-based (a) text editing and (b) text decoding. The white arrow means decoding.

MONET ($n$ times inference of decoder for $n$ objects), for an image with any number of objects, VCT only needs to inference one time.

**Objects & Properties Decomposition** Since the proposed requirements of concept tokens are both applicable for object-level representation and property factor variation, what information VCT encodes is data-driven. To verify this, we experiment on Objects-Room. As Figure 3 shows, visual tokens represent both object and property (background color and floor color). In contrast, as shown in Appendix C, COMET has difficulty learning factors like background colors.

### 4.4 Language Aligned Disentanglement

As we stated in Section 1, VCT tokenizes the visual concepts, like tokenization in NLP. In order to demonstrate the benefit of such tokenization. In this section, we utilize the pre-trained CLIP model to present such goodness of our method. Specifically, we use the pre-trained CLIP image encoder as the image tokenizer. Our experiments were mainly conducted on Shapes3D.

**CLIP Text Encoder-based Editing** CLIP aligns the text and image information. Therefore, the text encoder is also disentangled when we disentangle the image encoder. In order to verify this, we input a text prompt to derive concept tokens and replace the corresponding token of a image. Then we can edit the image by decoding. From Figure 6 (a), we see that the images are edited accordingly.

**CLIP Text Encoder-based Decoding** Since language is concept-level, like tokenizing a sentence (partitioning a sentence to words), VCT partitions information into individual concepts. Therefore, VCT transforms images into tokens similar to NLP. With VCT, we use the text encoder and VCT to decode the text prompt into an image. As shown in Figure 6 (b), VCT successfully differentiates the concept of ball/cube/cylinder (row 1) and small/None/big (row 2). See details in Appendix C.

## 5 Conclusion

In this paper, we demonstrate a general visual concept learning framework VCT to encode data into a set of tokens, which is a unified architecture to tackle disentangled representation learning and scene decomposition. We propose an attention-based tokenizer for the induction of information and a disentangling loss for encouraging the exclusivity of different tokens. VCT can be deployed to the intermediate representation to learn visual concepts. Although VCT is unsupervised, and the information encoded is data-driven, little hyper-parameter tuning is needed. Currently, we verify the effectiveness of VCT on small datasets. In future, we would like to scale up VCT to learn plentiful visual concepts in large-scale datasets. The potential negative societal impacts are malicious uses.

## Acknowledgement

We thank all the anonymous reviewers for their valuable comments. The work was supported in part with the National Natural Science Foundation of China (Grant No. 62088102).

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
