# Visual Concepts Tokenization
## *Appendix*

**Tao Yang**[1][*], **Yuwang Wang**[2][†], **Yan Lu**[2] , **Nanning Zheng**[1]
yt14212@stu.xjtu.edu.cn,
{yuwwan,yanlu}@microsoft.com,
nnzheng@mail.xjtu.edu.cn
[1]Xi'an Jiaotong University, [2]Microsoft Research Asia

## A  Implementation Details & More Quantitative Results

For the setting of the baselines of disentangled representation learning, we follow DisCo [9]. For the setting of COMET, we follow their own setting for global factors.

For the setting of the baselines of scene decomposition, we follow their own settings. Specifically, for the setting of COMET, we follow their own setting for local factors (Clevr). As for slot attention, we also follow their setting.

### A.1  Setting for VCT

We empirically set the batch size to 32, the coefficient of disentanglement term $\lambda_{dis}$ to 1, the layer number of Concept Tokenizer $L_E$ to 6, the layer number of Concept Detokenizer $L_D$ to 4, and feature dimension $D$ to 256 for all the datasets. In addition, we set the number of image tokens $N$ to 256 for an image size of $64 \times 64$, and the number of concept tokens $M$ to 20 for both two tasks. In the training of VCT, we use an Adam optimizer [5], set the learning rate to $1e-4$, and keep other parameters as the default setting of PyTorch [8], train 80k iterations for convergence. Due to the usage of pre-trained VQ-VAE as image tokenizer and detokenizer, VCT converges faster than random initialized VCT. Consider that VQ-VAE is pre-trained, and $\mathcal{L}_{dis}$ is optimized only w.r.t. the Concept Tokenizer, VCT only consumes about 10G GPU memories. Therefore, we only use one Tesla P100 16G GPU for training.

Since the feature dimension of the CLIP image encoder is 512, when we adopt it as the image tokenizer, we use a set of $N$ MLPs to map the CLIP feature into $N$ image tokens, respectively. Besides, CLIP has no decoder. We use a network that with the same architecture of VQ-VAE decoder as the image detokenizer. The pre-trained CLIP model is from the official implementation[3].

### A.2  Architecture of VQ-VAE

The VQ-VAE we adopted is from the PyTorch implementation[4]. We follow [10] to train VQ-VAE 200 epochs for each dataset. The encoder and decoder architectures are same to [10], which is shown in Tables A.2 and A.2. For the training hyper-parameters, we follow VQ-VAE to set.

---

[*]Work done during internships at Microsoft Research Asia.

[†]Corresponding author

[3]https://github.com/openai/CLIP

[4]https://github.com/nadavbh12/VQ-VAE

36th Conference on Neural Information Processing Systems (NeurIPS 2022).

| |
|---|
| Conv 256, $4 \times 4$, `stride` $= 2$, `padding` $= 1$ |
| BatchNorm |
| ReLu |
| Conv 256, $4 \times 4$, `stride` $= 2$, `padding` $= 1$ |
| BatchNorm |
| ReLu |
| ResBlock(256, 256) |
| BatchNorm |
| ResBlock(256, 256) |
| BatchNorm |

Table 1: Encoder architecture of VQ-VAE used in VCT.

| |
|---|
| ResBlock(256, 256) |
| BatchNorm |
| ResBlock(256, 256) |
| BatchNorm |
| ConvTranspose 256, $4 \times 4$, `stride` $= 2$, `padding` $= 1$ |
| BatchNorm |
| ReLu |
| ConvTranspose 256, $4 \times 4$, `stride` $= 2$, `padding` $= 1$ |

Table 2: Decoder architecture of VQ-VAE used in VCT.

## A.3 Settings for Ablation Study

For the "AE" tokenizer/deTokenizer, we adopt autoencoder with the same architecture to the VQ-VAE and use them as image tokenizer/deTokenizer directly. In addition, we use the MSE loss for reconstruction. For the "Patch" tokenizer/deTokenizer, we use a Conv layer with patch size set to $4$ and channels set to $256$. We also adopt a standard MSE loss as $\mathcal{L}_{rec}$.

For the "CNN DeTokenier", we adopt a CNN structure as shown in Table A.3. We first reshape the concept tokens to $B \times N \times 16 \times 16$, where $B$ is batch size. Then we decode the image tokens by a CNN. For the transformer detokenizer, We first append $N - M$ learnable tokens to the encoded concept tokens, then decode them to image tokens by four layers of self-attention blocks (self attention with FFN and layer norm), which is the same as the self-attention block in Concept Tokenizer.

| |
|---|
| ConvTranspose 16, $3 \times 3$, `stride` $= 1$, `padding` $= 1$ |
| ReLu |
| ConvTranspose 32, $3 \times 3$, `stride` $= 1$, `padding` $= 1$ |
| ReLu |
| ConvTranspose 64, $3 \times 3$, `stride` $= 1$, `padding` $= 1$ |
| ReLu |
| ConvTranspose 128, $3 \times 3$, `stride` $= 1$, `padding` $= 1$ |
| ReLu |
| Linear 256 |

Table 3: Architecture of CNN Detokenizer.

## A.4 More Quantitative Results

The results of disentangled representation learning on the MIG and BetaVAE disentanglement metrics is presented in Table 4.

| Method | Cars3D | | Shapes3D | | MPI3D | |
|---|---|---|---|---|---|---|
| | MIG | BetaVAE | MIG | BetaVAE | MIG | BetaVAE |
| *VAE-based:* | | | | | | |
| FactorVAE | $0.142 \pm 0.023$ | $1.00 \pm 0.000$ | $0.434 \pm 0.143$ | $0.892 \pm 0.064$ | $0.099 \pm 0.029$ | $0.348 \pm 0.012$ |
| $\beta$-TCVAE | $0.080 \pm 0.023$ | $0.999 \pm 1.0e-4$ | $0.406 \pm 0.175$ | $0.978 \pm 0.036$ | $0.114 \pm 0.042$ | $0.339 \pm 0.029$ |
| *GAN-based:* | | | | | | |
| InfoGAN-CR | $0.011 \pm 0.009$ | $0.450 \pm 0.022$ | $0.297 \pm 0.124$ | $0.837 \pm 0.039$ | $0.163 \pm 0.076$ | $0.450 \pm 0.022$ |
| *Pre-trained GAN based:* | | | | | | |
| LD | $0.086 \pm 0.029$ | $0.999 \pm 2.54e-4$ | $0.168 \pm 0.056$ | $0.913 \pm 0.063$ | $0.097 \pm 0.057$ | $0.535 \pm 0.057$ |
| CF | $0.083 \pm 0.024$ | $1.000 \pm 0.000$ | $0.307 \pm 0.124$ | $0.999 \pm 0.001$ | $0.183 \pm 0.081$ | $0.669 \pm 0.033$ |
| GS | $0.136 \pm 0.006$ | $1.000 \pm 0.000$ | $0.121 \pm 0.048$ | $0.944 \pm 0.044$ | $0.163 \pm 0.065$ | $0.605 \pm 0.061$ |
| DS | $0.118 \pm 0.044$ | $1.000 \pm 0.000$ | $0.356 \pm 0.090$ | $0.991 \pm 0.022$ | $0.093 \pm 0.035$ | $0.651 \pm 0.043$ |
| DisCo | $0.179 \pm 0.037$ | $0.999 \pm 6.86e-5$ | $0.512 \pm 0.068$ | $0.987 \pm 0.028$ | $0.222 \pm 0.027$ | $0.530 \pm 0.015$ |
| *Concept-based:* | | | | | | |
| COMET | $0.000 \pm 0.000$ | $0.343 \pm 0.006$ | $0.0002 \pm 0.000$ | $0.166 \pm 0.004$ | $0.000 \pm 0.0001$ | $0.144 \pm 0.005$ |
| VCT (Ours) | $0.117 \pm 0.045$ | $\mathbf{1.00 \pm 0.000}$ | $\mathbf{0.525 \pm 0.028}$ | $\mathbf{0.999 \pm 0.0004}$ | $\mathbf{0.227 \pm 0.048}$ | $\mathbf{0.844 \pm 0.038}$ |

Table 4: Comparisons of disentanglement on the MIG and BetaVAE disentanglement metrics (mean $\pm$ std, higher is better). VCT achieves the state of the art performance with a large margin in almost all the cases compared to all of the baselines. Especially on the MPI3D dataset.

# B  More Qualitative Results

## B.1  Qualitative Results on Shapes3D

**Identify the meaningful tokens** As shown from Figure 1, the meaningful concept tokens emerge randomly. Intuitively, if the meaningless tokens are different from each other, VCT can recognize them easily when optimizing $\mathcal{L}_{dis}$. Therefore, the meaningless tokens should have a much smaller variance than meaningful ones. In order to identify these meaningful concepts, we can calculate the variance of the tokens across a large batch covering all variations. The concept tokens with high variance are meaningful ones. Since the token is vector-wise, the variance is also a vector. Therefore, we calculate $\ell_2$ norm of each variance vector. As shown in Figure 2, the concept tokens with the norm of variance vector larger than 0.1 are meaningful ones. e.g., 7, 8, 14, 15, 19, 20.

**Linear interpolations** In order to analyze the linearity of the concept token space. We linearly interpolate the concepts token $c_i^k$ and $c_j^k$ by equation $c^k = \alpha c_i^k + (1 - \alpha)c_j^k$, where $\alpha \in (0, 1)$. Interestingly, as shown in Figure 3, the space of concept tokens is close to linear space.

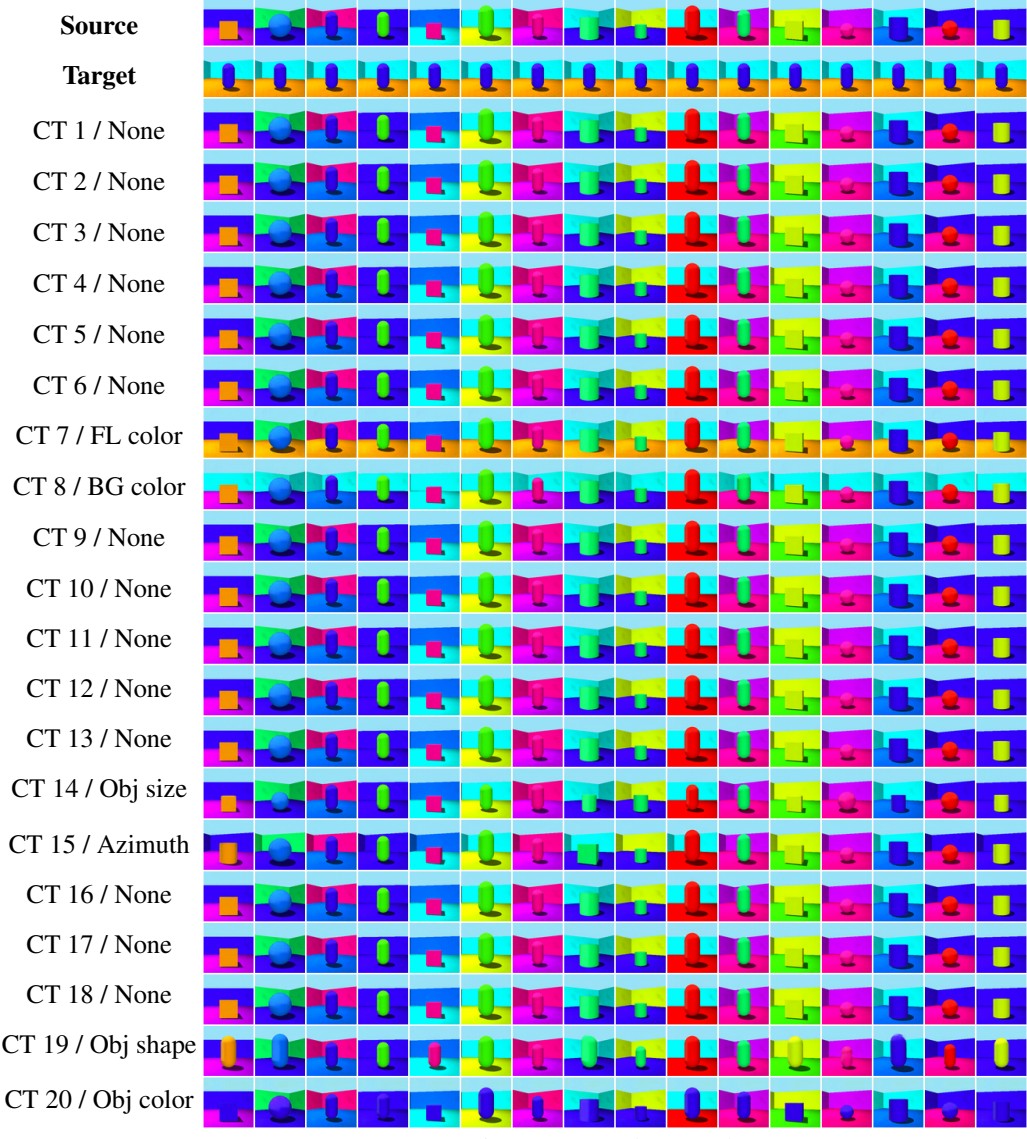

Swapping concept token on Shapes3D

Figure 1: Visualization of swapping concepts on Shapes3D. The images in the first row provide source concepts, and the second provides target concepts. The rest of the images are swapped ones ("CT i" represent that the row is corresponding to $i$-th concept token swapped image. "None" denotes meaningless concept; "Obj" denotes Object; "FL" denotes floor; "BG" denotes background).

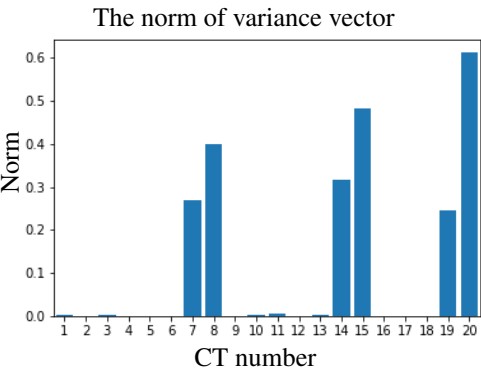

Figure 2: Variance vector norm of concept tokens. We calculate the variance of concept tokens across a batch of instances and obtain a variance vector for each concept. Then, we calculate $\ell_2$ norm of the variance vector.

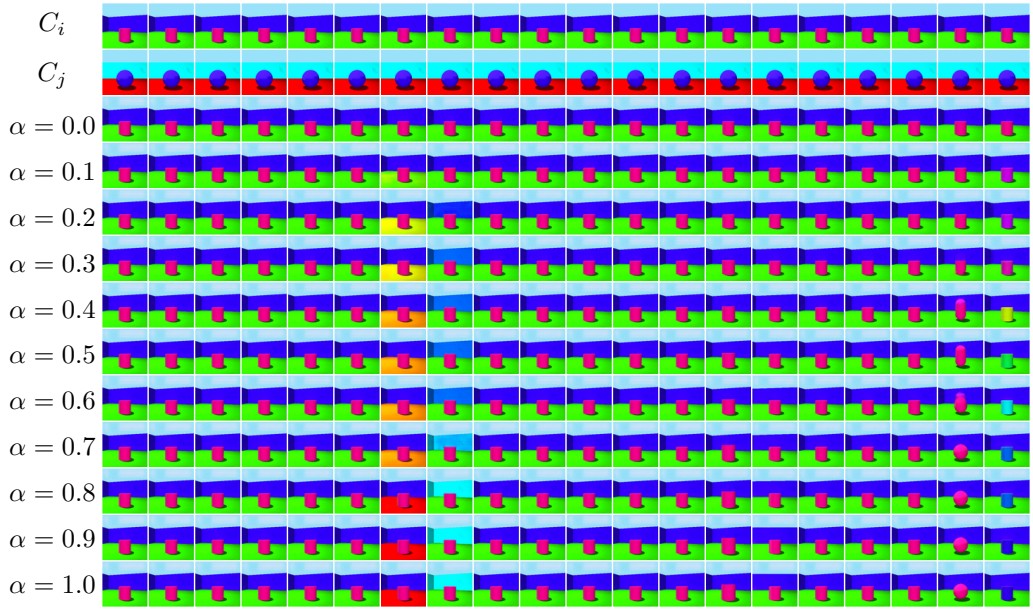

Figure 3: Visualization of linear interpolation of concept tokens on Shapes3D. The images in the first row provide source concepts, and the second provides the target concept. The rest of the images are interpolated ones (each column represents a single concept interpolation).

## B.2 Qualitative Results on MPI3D

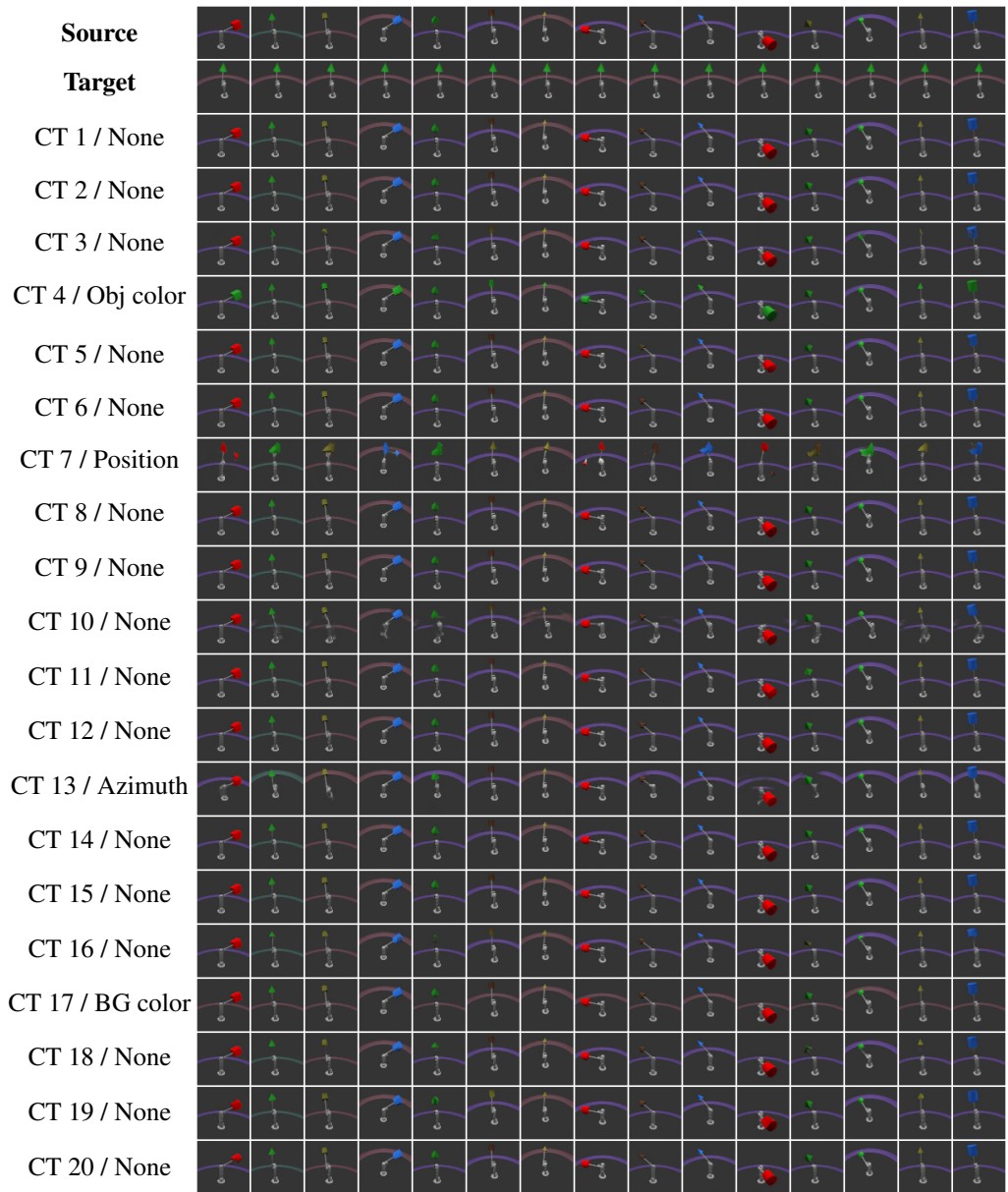

Swapping concept token on MPI3D

Figure 4: Visualization of swapping concepts on MPI3D. The images in the first row provide source concepts, and the second provides the target concept. The rest of the images are swapped ones ("CT i" represents that the $i$-th concept token of the source image is replaced with the one of target image. "None" denotes meaningless concept; "Obj" denotes Object; "BG" denotes background).

## B.3 Qualitative Results on Car3D

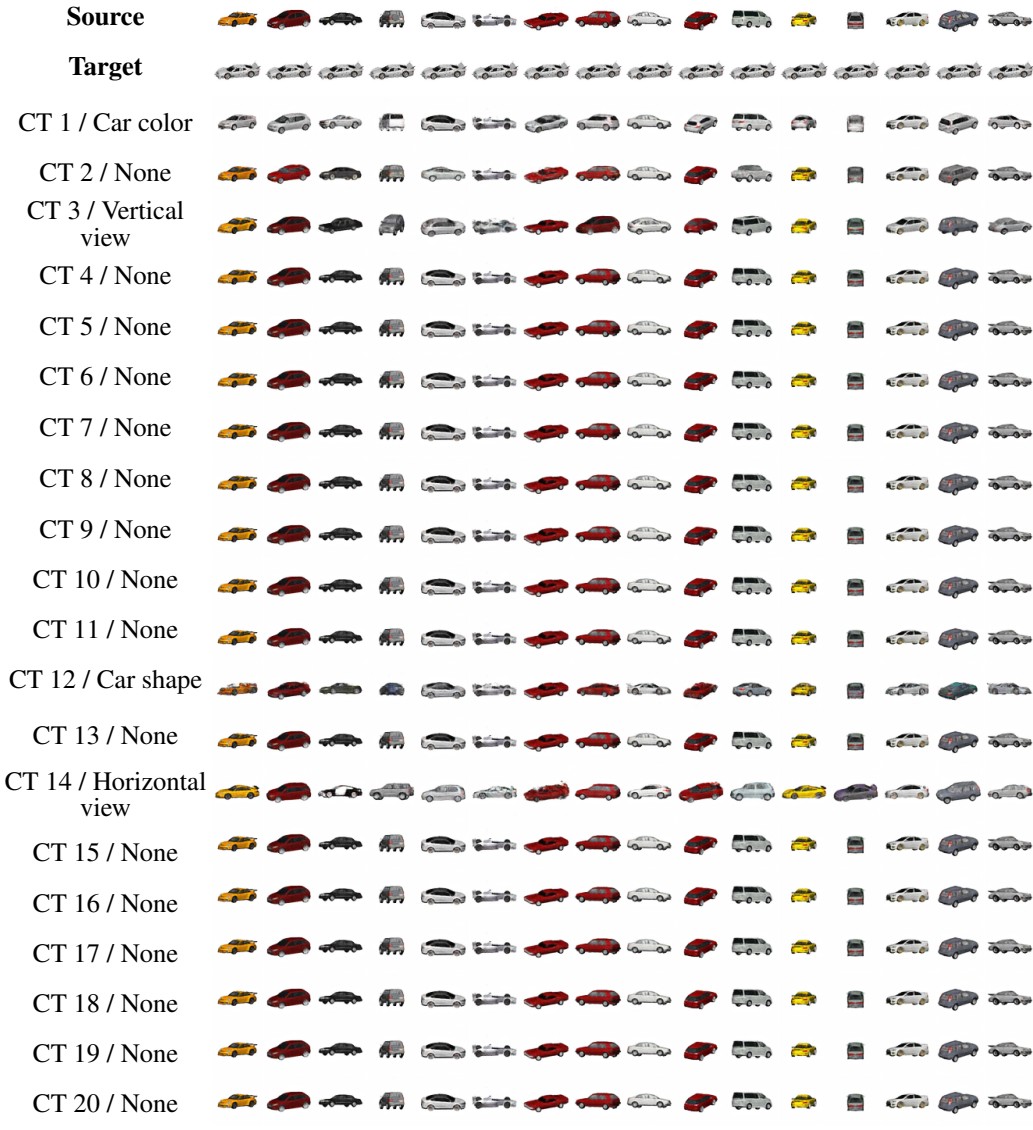

Swapping concept token on Car3D

Figure 5: Visualization of swapping concepts on Car3D. The images in the first row provide source concepts, and the second provides the target concept. The rest of the images are swapped ones ("CT i" represent that the row is corresponding to $i$-th concept token swapped image. "None" denotes meaningless concept).

## B.4 Qualitative Results on CeleBA

**Source**

**Target**

CT 1 / Concept 1

CT 2 / Right Collar

CT 3 / Cheeks

CT 4 / Left Collar

CT 5 / Bangs

CT 6 / None

CT 7 / None

CT 8 / None

CT 9 / None

CT 10 / None

CT 11 / None

CT 12 / None

CT 13 / None

CT 14 / None

CT 15 / None

CT 16 / None

CT 17 / None

CT 18 / None

CT 19 / Mouth

CT 20 / Eyes & Nose

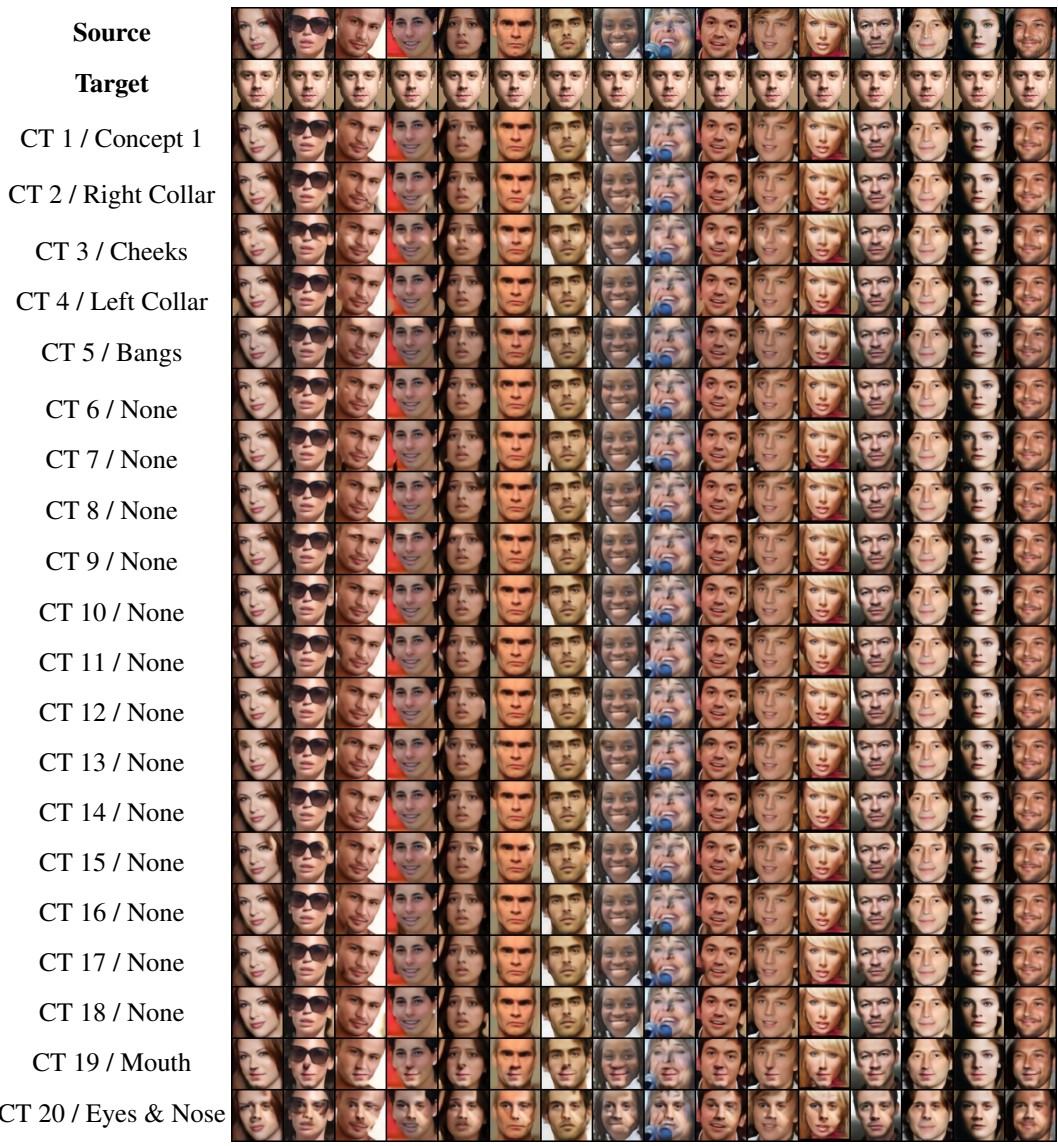

Swapping concept token on CeleBA

Figure 6: Visualization of swapping concepts on CeleBA. The images in the first row provide source concepts, and the second provides the target concept. The rest of the images are swapped ones ("CT i" represent that the row is corresponding to $i$-th concept token swapped image. "None" denotes meaningless concept).

## B.5 Results on Real-world Datasets

In this section, we demonstrate that VCT is also effective (i.e., generalizes well) on real-world images. We train VCT on two challenging real-world datasets MSCOCO (resolution: $224 \times 224$, concept tokens number: 20) [6] and KITTI (resolution: $64 \times 64$, concept tokens number: 20) [3], and conduct qualitative evaluations. To the best of our knowledge, we are the first to conduct unsupervised disentangled representation learning on MSCOCO and KITTI.

VCT can be flexibly combined with different architectures. For MSCOCO, we take the encoder and decoder of BEiT [1] as the image tokenizer/detokenizer for VCT to leverage a large pretraind vision model. As for KITTI, we still utilize pretrained VQ-VAE as the pretrained vision model.

As shown in Figure 7, the results on MSCOCO and KITTI demonstrate that VCT is able to learn the concepts of the sky, ground, photos, and chairs. This is quite similar to what VCT can learn on the synthesized dataset Objects-Room. As the real-world dataset is more diverse, we observe several failure cases shown in Figure 8. We suppose those failure cases are due to VCT, trained with reconstruction loss, is not good at synthesizing counterfactual samples which are far from the data distribution. We also show the results on KITTI in Figure 9. The results show that VCT can extract visual concepts such as shadow, sky, car, house, ground, etc. Due to the time and computation limitation, we test VCT on a low resolution.

In summary, the results on MSCOCO and KITTI demonstrate that VCT is also effective in real-world scenarios. Meanwhile, the quality is not as good as in the synthesized datasets. Note that disentanglement in the real world is still quite challenging. Compared to synthesized data, the real-world data contains more diverse and unlimited scene variations, the total number of concepts is large and unknown, and the number of concepts is image specific. Compared to the street dataset KITTI, MSCOCO has more diverse visual contents and image variations, and it is harder to learn visual concepts.

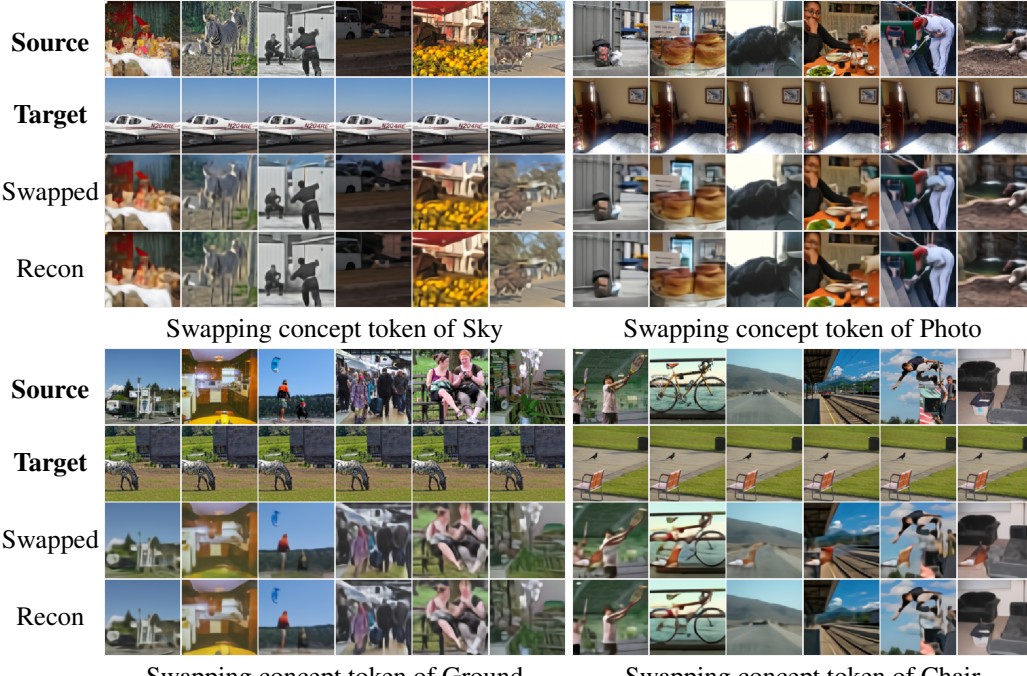

Figure 7: Visualization of swapping the meaningful concepts on MSCOCO. The images in the first row provide source concepts, and the second provides the target concept. The rest of the images are swapped and reconstructed ones("Recon" represents that the row corresponds to the reconstructed image).

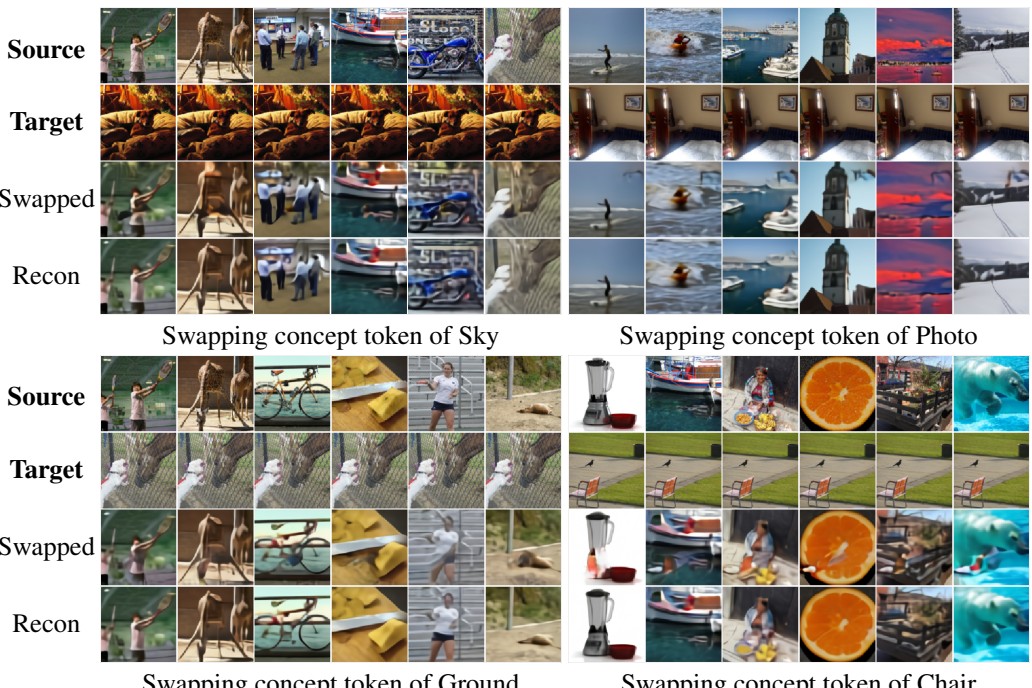

Figure 8: Visualization of **the failed cases** of swapping concepts on MSCOCO. The images in the first row provide source concepts, and the second provides the target concept. The rest of the images are swapped and reconstructed ones ("Recon" represent that the row corresponds to the reconstructed image).

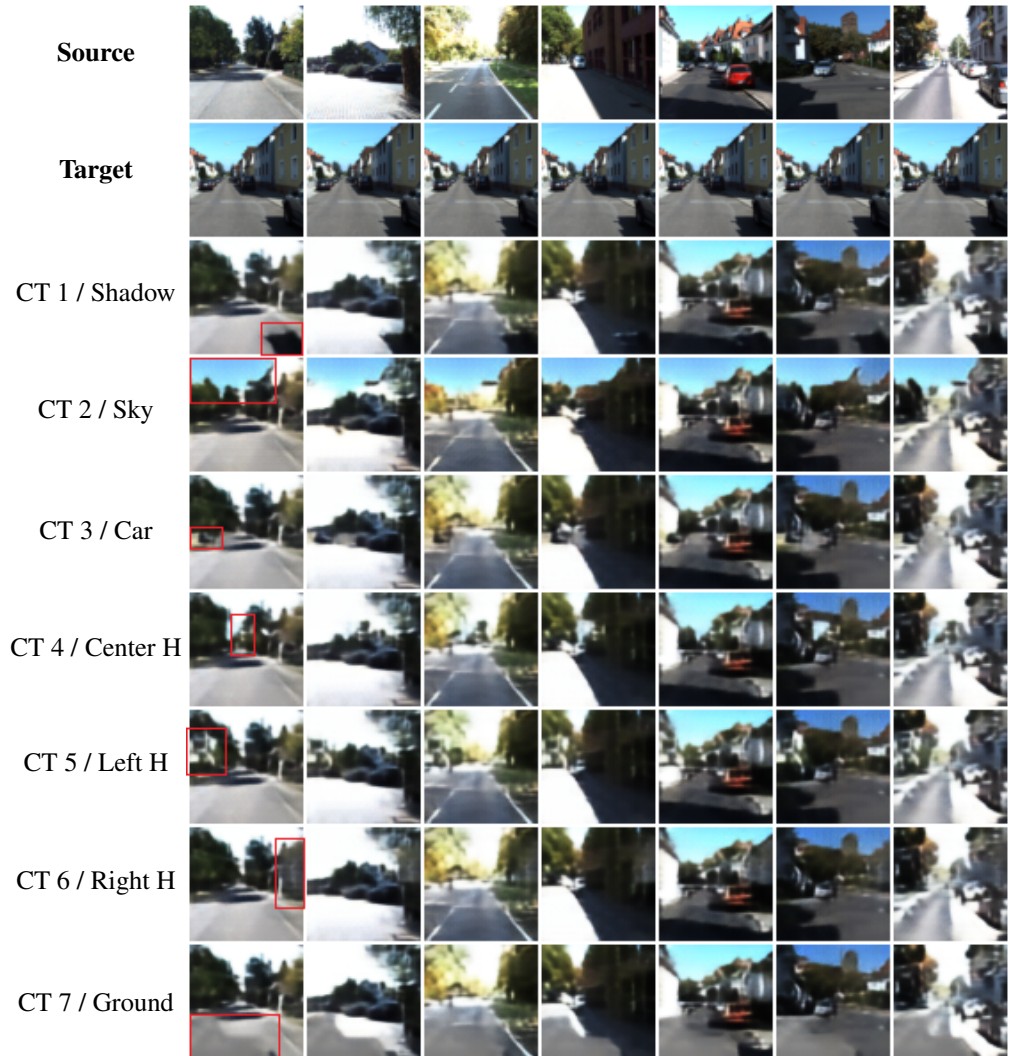

Swapping concept token on KITTI

Figure 9: Visualization of swapping the meaningful concepts on KITTI. The images in the first row provide source concepts, and the second provides the target concept. The rest of the images are swapped ones ("CT i" represent that the row is corresponding to $i$-th concept token swapped image, we use "H" to represent "house" in short).

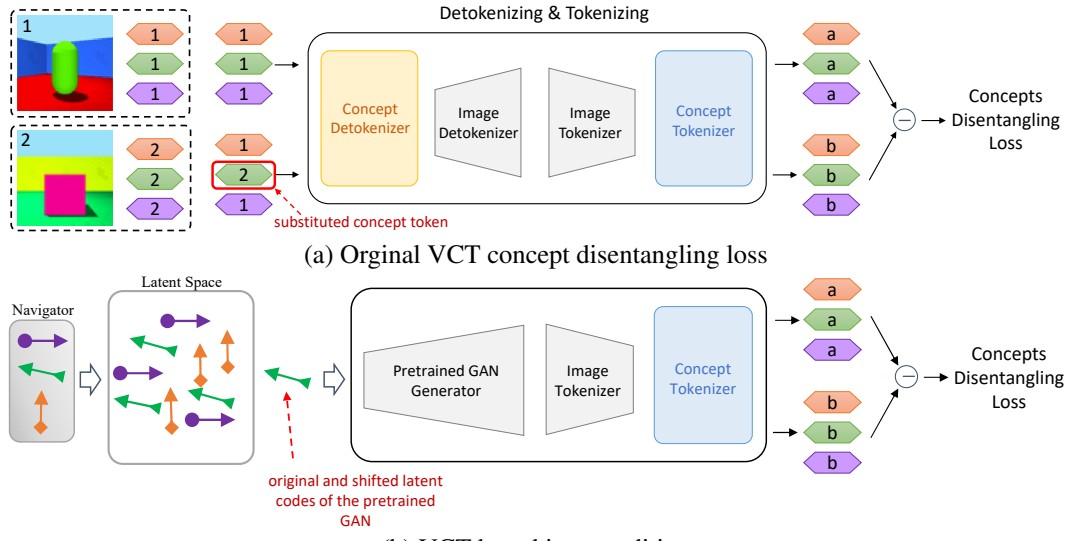

(a) Orginal VCT concept disentangling loss

(b) VCT-based image editing

Figure 10: Difference between proposed VCT for editing and original VCT. We abnegate the reconstruction loss and use latent code shift to substitute token swapping. We add a navigator to provide learnable directions. The figure of the navigator is taken from DisCO [9]. Similar to DisCo, we adopt a CNN encoder as the Image Tokenizer.

Furthermore, inspired by DisCO [9], we present a new architecture by combining the tokenizer of VCT with a pretrained GAN for image editing. As shown in Figure 10, this new architecture takes a pretrained GAN as the concept and image detokenizer and uses a navigator for discovering semantic directions which correspond to these concepts. As we adopt a pretrained GAN and keep it fixed, the reconstruction loss can be discarded. For concept disentangling loss, we use latent code shift to substitute token swapping operation. In order to verify the effectiveness of this new architecture, we train this architecture on several real-world datasets. For Cats/Church/FFHQ (resolution: $256 \times 256$, concept tokens number: 32), the pretrained GAN is StyleGAN2 [4]. For ImageNet (resolution: $256 \times 256$, concept tokens number: 32), the pretrained GAN is BigGAN [2]. The results are shown in Figure 11(Cats), Figure 12 (Church),Figure 14 (FFHQ),Figure 13 (ImageNet). Generally, VCT can discover many disentangled concepts via discovering the directions in the latent space of pretrained GANs. In this way, VCT can be used for image editing.

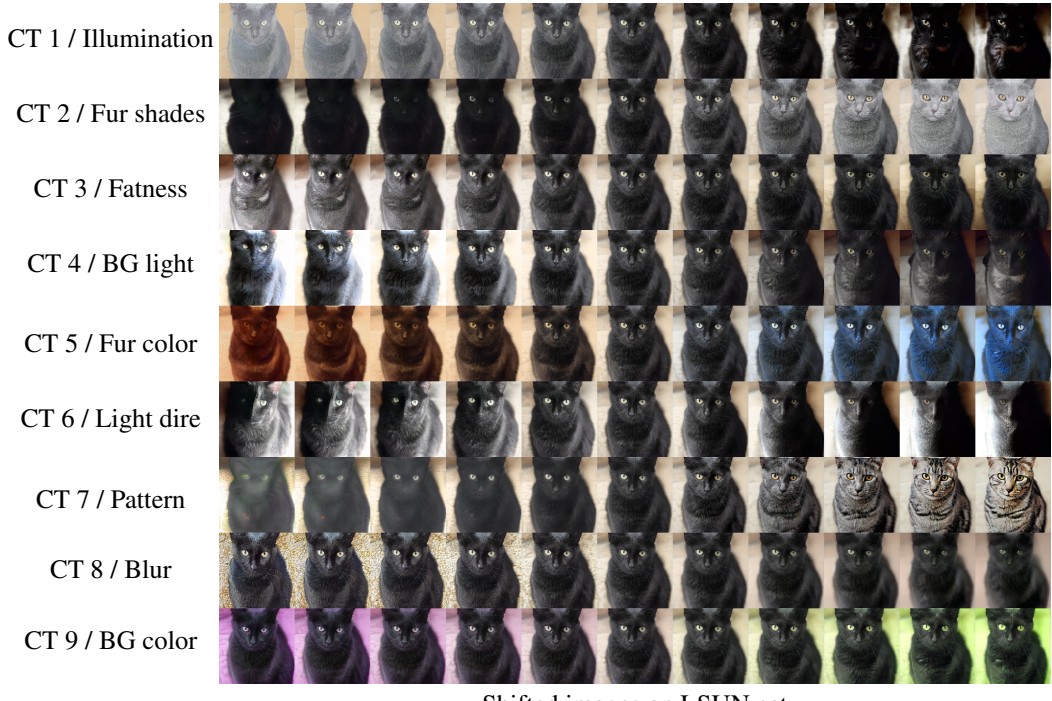

CT 1 / Illumination

CT 2 / Fur shades

CT 3 / Fatness

CT 4 / BG light

CT 5 / Fur color

CT 6 / Light dire

CT 7 / Pattern

CT 8 / Blur

CT 9 / BG color

Shifted images on LSUN cat

Figure 11: Visualization of (**VCT for image editing**) shifting latent code along the meaningful directions of corresponding concept token on LSUN cat. The 6-th image in each row is the source image, and the rest of the images are shifted ones ("CT i" represent that the row is corresponding to latent code shifted images of corresponding concept token, we use "BG" to represent the background and "dire" to represent the direction in short).

CT 1 / Sky color

CT 2 / Cloudy

CT 3 / H color

CT 4 / Grass

CT 5 / Vitality

CT 6 / Season

CT 7 / Sky blue

CT 8 / fur shades

CT 9 / Lake

CT 10 / Rainy

CT 11 / H size

CT 12 / H material

CT 13 / Spire

CT 14 / H shades

CT 15 / Sharpness

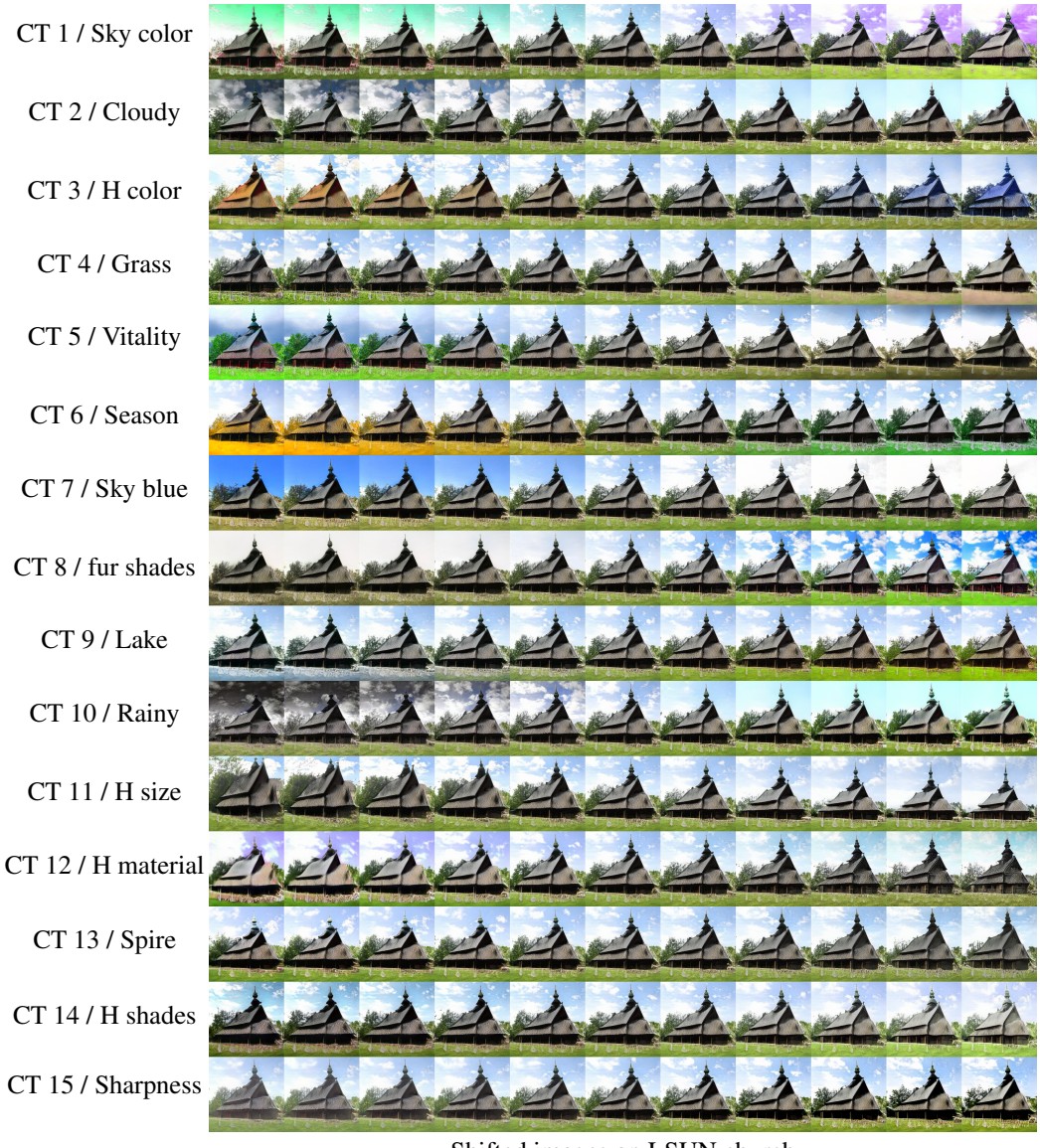

Shifted images on LSUN church

Figure 12: Visualization of (**VCT for image editing**) shifting latent code along the meaningful directions of corresponding concept token on LSUN church. The 6-th image in each row is the source image, and the rest of the images are shifted ones ("CT i" represent that the row is corresponding to latent code shifted images of corresponding concept token, we use "BG" to represent background in short).

CT 1 / Standing

CT 2 / Azimuth

CT 3 / Zoom in

CT 4 / Dog size

CT 5 / White

CT 6 / FL color

CT 7 / FL size

CT 8 / BG

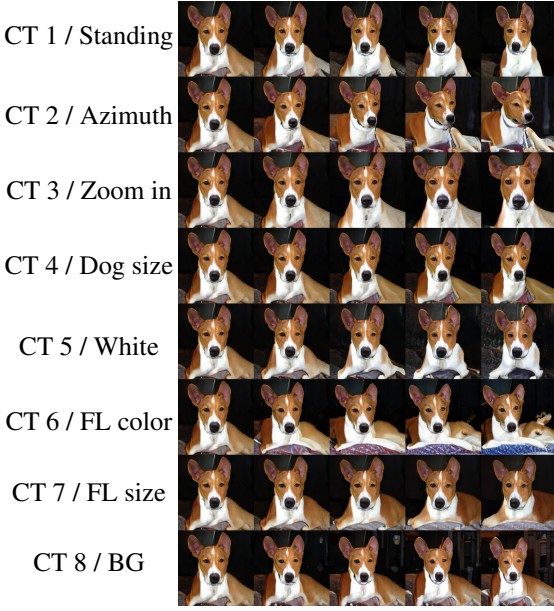

Shifted images on ImageNet dog

Figure 13: Visualization of (**VCT for image editing**) shifting latent code along the meaningful directions of corresponding concept token on ImageNet dog class. The 6-th image in each row is the source image, and the rest of the images are shifted ones ("CT i" represent that the row is corresponding to latent code shifted images of corresponding concept token, we use "BG" to represent background and "FL" to represent floor in short).

CT 1 / Hair color

CT 2 / color

CT 3 / Smile

CT 4 / Skin color

CT 5 / Make up

CT 6 / Bald head

CT 7 / Gender

CT 8 / white hair

CT 9 / Eyes

CT 10 / Bangs

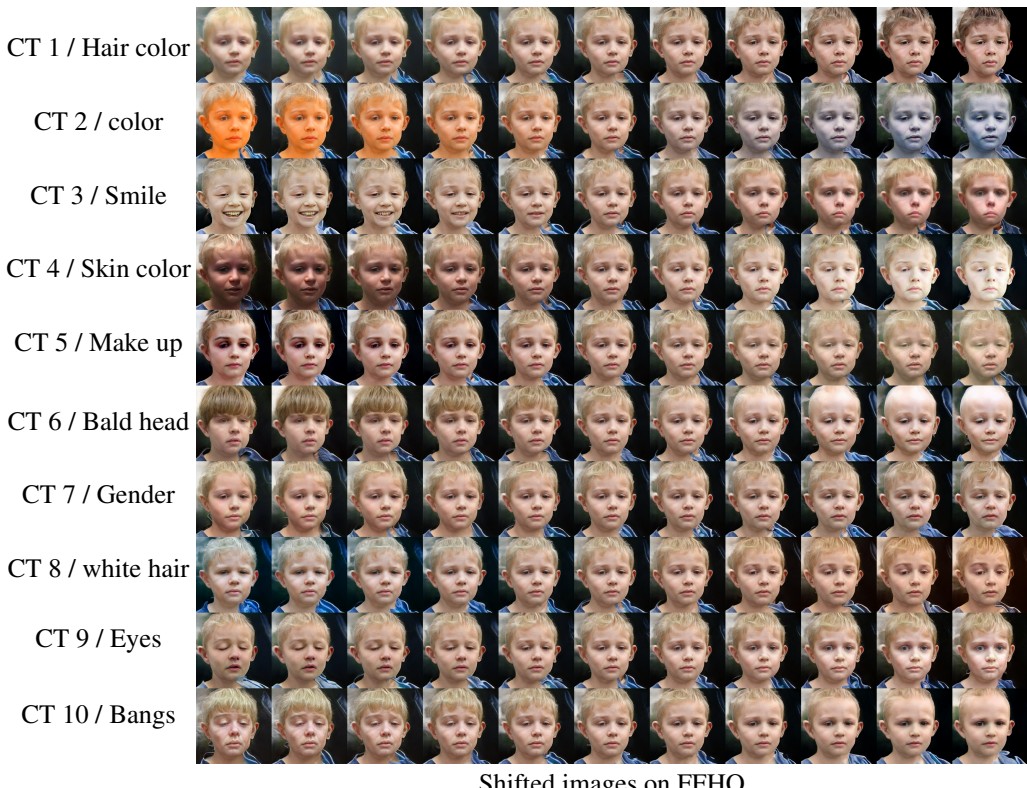

Shifted images on FFHQ

Figure 14: Visualization of (**VCT for image editing**) shifting latent code along the meaningful directions of corresponding concept token on FFHQ. The 6-th image in each row is the source image, and the rest of the images are shifted ones ("CT i" represents that the row is corresponding to latent code shifted images of corresponding concept token).

Table 5: Sensitivity of VCT on batchsize and token numbers.

| Setting | MIG | DCI |
|---|---|---|
| batchsize = 3 | 0.418 | 0.790 |
| batchsize = 16 | 0.497 | 0.862 |
| batchsize = 32 | 0.525 | 0.884 |
| batchsize = 64 | **0.535** | **0.900** |
| tokens number = 3 | 0.450 | 0.599 |
| tokens number = 10 | **0.533** | 0.867 |
| tokens number = 20 | 0.525 | 0.884 |
| tokens number = 30 | 0.493 | **0.885** |

## B.6 Are the sensitivity results (batch sizes, token numbers) dependent on the datasets?

For a dataset of more complex scenarios (i.e., with more GT concepts), a larger token number $M$ is needed in VCT. Specifically, $M$ should be no smaller than the number of GT concepts. Since we apply the disentangling loss inside each batch, to ensure the diversity inside a batch, the batch size should also be no smaller than the number of GT concepts. As the number of GT concepts is usually relatively small in the synthesized data, e.g., 6 for Shape3D, the setting of $M$ and batch size are often satisfied. Furthermore, in order to verify this, we add an experiment on Shapes3D with the token number $M$ and batch size set to 3, which are smaller than the number of the GT factors/concepts number 6. As the table shows below, the performance significantly drops. However, if the concept tokens number is already $\geq$ GT factors/concepts number, the performance is robust to the concept tokens number (See Table 5). Therefore, under the condition that token number/ batch size $\geq$ GT factors/concepts number.

## C Supplement on Scene decompsoition & Languagealigned Disentanglement

### C.1 Decoder for Explicit Mask

For the image detokenizer, we add an extra channel in output to predict the explicit mask for each concept. We follow slot attention [7] to decode objects once a time and combine them with explicit masks. Specifically, since we use cross-attention in the Concept Detokenizer, the number of decoded image tokens with a single concept token as input is the same as the ones with all concept tokens as input. Therefore, We decode a single concept token into image tokens and decode image tokens to the image and mask using the image detokenizer. In this way, we can decode these concept tokens $\{C_i^j\}$ into images $x_{ij}$ and masks $m_{ij}$ (after softmax across masks), then combine them with the following equation

$$x_i = \sum_j^M x_{ij} m_{ij} \tag{1}$$

### C.2 Decomposition and Recombination

In this section, we provide the details of using VCT to achieve scene decomposition and recombination.

Given a scene image, VCT represents a single object inside the scene with a single concept token for scene decomposition. Therefore, we can spatially decompose the scene image into objects. We first identify the "background token" inside the dataset. Here, we swap tokens between images and locate the "background token" as the token that results in nearly no difference on the decoded image. We find that the "background token" is not image-specific. Given a scene image to decompose, we get its concept tokens via VCT, then we only keep one object token and replace other tokens with "background token", then decode them to an image. In this way, we can get an image only with this object. Similarly, we can get a set of images, and each image contains a single object. Besides, we can also replace object tokens one by one to add an object once a time, as shown in Figure 4 and 5 (c) in the main paper.

With such decomposition ability, we can recombine different objects in different scenes. Given two images: image A and image B, by replacing tokens of image A with object tokens in image B, we can add the objects of image B to image A (no overlapping in spatial). As shown in Figure 5 in the main paper. If there is overlap between the two objects, the replacing operation will result in replacing them.

### C.3 Language-aligned Disentanglement

In this section, we demonstrate that VCT can achieve language-aligned disentanglement by combining with pretrained CLIP encoders, which can be applied for text-guided image editing or text-to-image generation. As for the architecture, we connect a pretrained CLIP image encoder with VCT via an MLP, which maps the output of the CLIP encoder to a set of tokens. After training VCT, we replace the pretrained CLIP image encoder with pretrained CLIP text encoder. Thus the model can take text as input and achieve text-guided image editing (replacing concept tokens of image and text and decoding) or text to image generation (decoding the concept tokens of text). The image editing results are shown in Figure 15(a) and text to image generation results are shown in Figure 15(b). Note that, only trained on images, VCT can successfully extract the visual concepts, which are also aligned with text concepts, so that we can control the generation process.

### C.4 COMET Results on Objects-Room

We use the default setting of Clevr in COMET and train it using the official implementation[5]. As figure 16 shows, none of the concepts can generate the background or the floor alone. In addition, we do not observe such global factors learned in COMET, as shown in Figure 18. there is no such energy function that has a high response on the background or floor.

---

[5]https://github.com/yilundu/comet

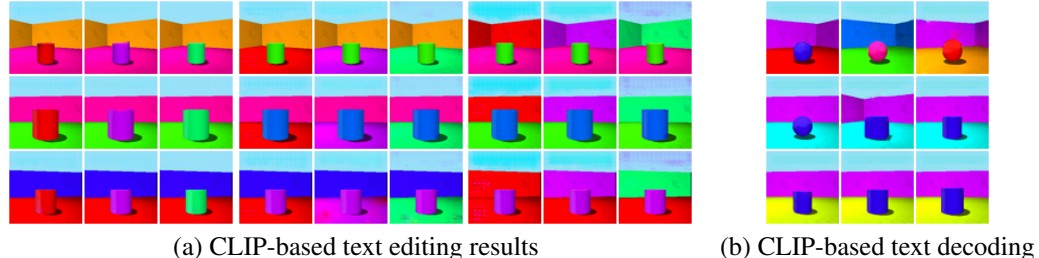

(a) CLIP-based text editing results          (b) CLIP-based text decoding

Figure 15: (a) The images can be controlled by replacing the concept tokens of the corresponding text. Left: " The object color is {red/purple/green}". ({red/purple/green} for the {1st/2nd/3rd} column of the images, respectively) Middle: " The floor color is {red/purple/green}". Right: "The background color is {red/purple/green}. (b) The decoded images when we take the corresponding text as input. First row text: "The background color is {purple/blue/purple}, and the object is a small blue/red/red ball, and the floor color is {red/green/orange}." Second-row text: "The background color is purple, and the object is a small blue {ball/cube/cylinder}, and the floor color is cyan". Third-row text: "The background color is purple, and the object is a {small/[empty]/large} blue cylinder, and the floor color is yellow".

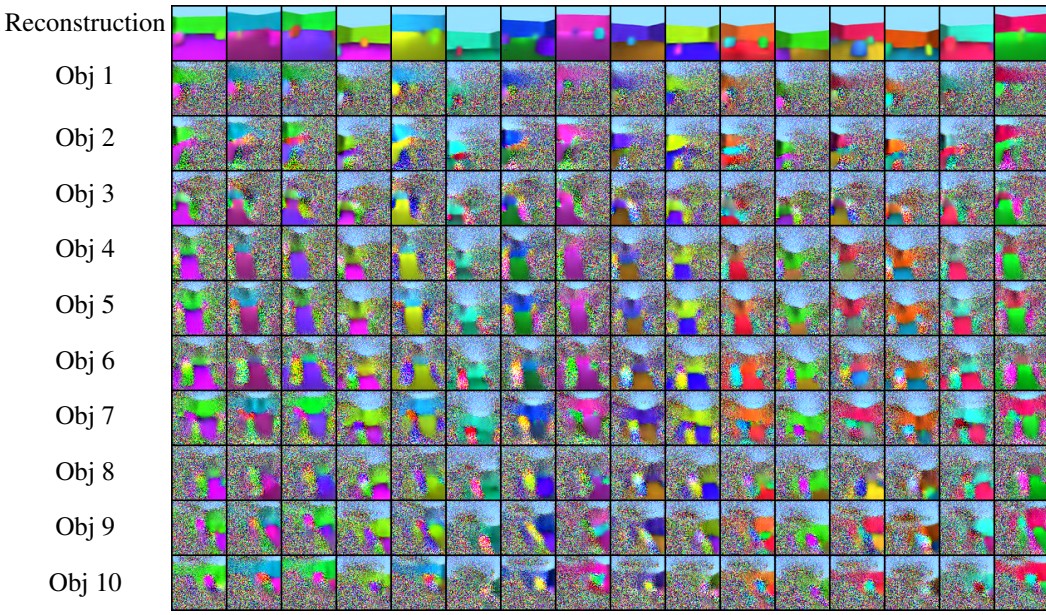

Single concept generation on Objects-Room

Figure 16: Visualization of single concept generation on Objects-Room. The images in the first row are the reconstructed images, and the following rows provide the image generation results of a single concept.

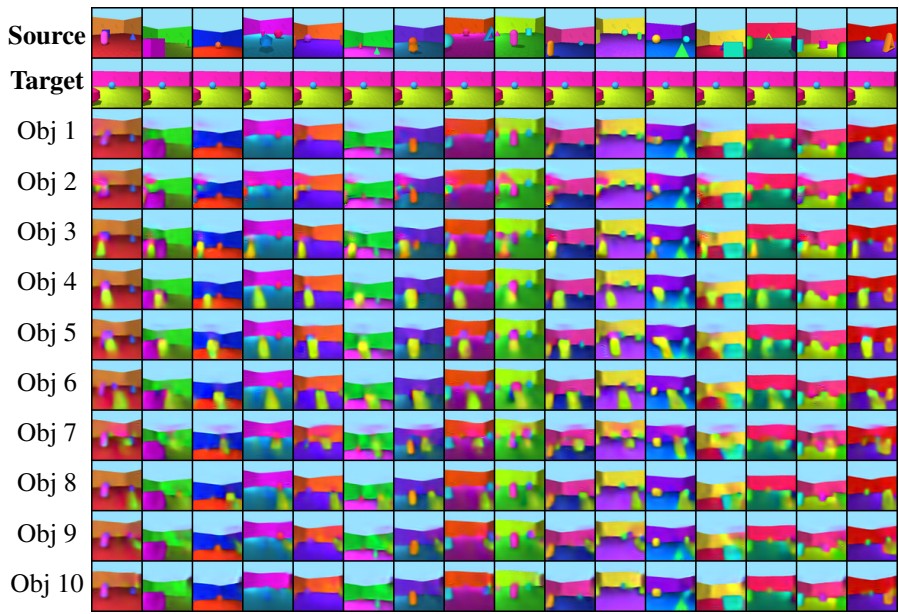

Swapping concept token of COMET on Object-Room

Figure 17: Visualization of swapping concept token of COMET on Object-Room. The images in the first row provide source concepts, and the second provides the target concept. The rest of the images are swapped ones

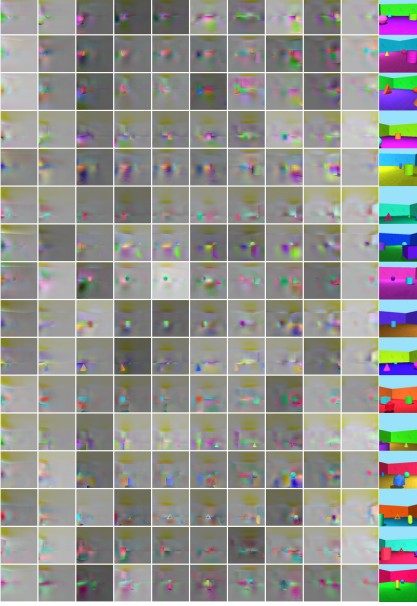

Figure 18: Visualization of the gradient of the energy function for each concept in Objects Room. The last column represents the data for computing gradient, and each column provides the gradient of the energy function of each concept.

## C.5 More Qualitative Results

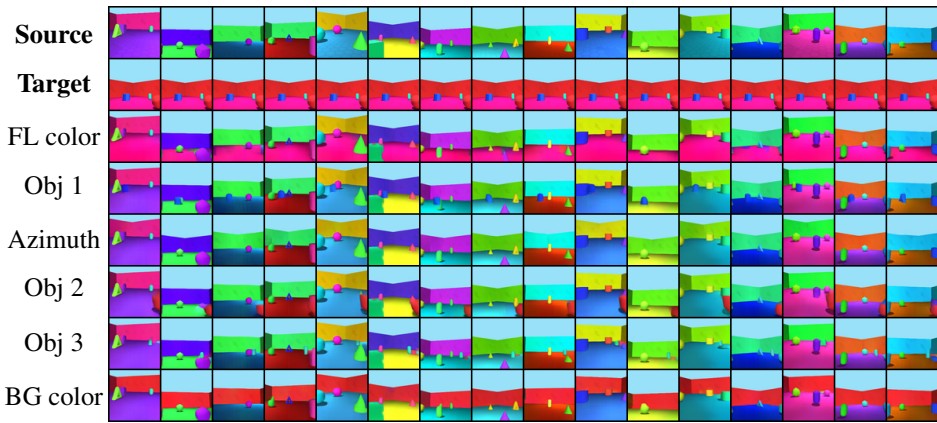

Swapping concept token on Objects-Room

Figure 19: Visualization of swapping concepts results on Objects-Room. The images in the first row provide source concepts, and the second provides the target concept. The rest of the images are swapped ones ("Obj" denotes Object; "FL" denotes floor; "BG" denotes background).

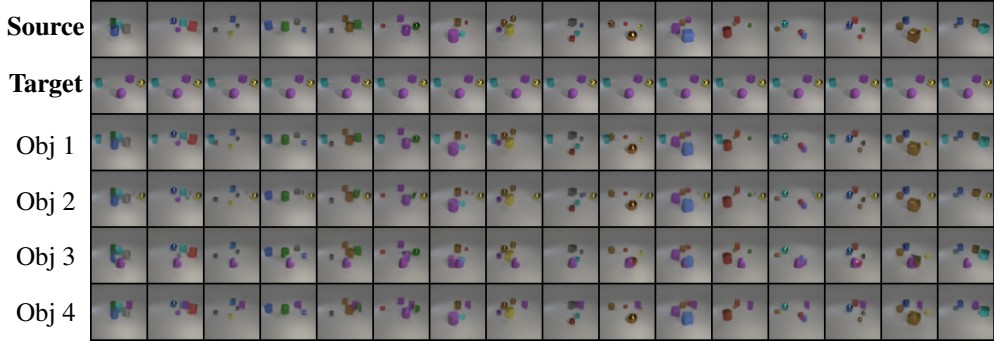

Swapping concept token on Clevr

Figure 20: Visualization of swapping concepts results on Clevr. The images in the first row provide source concepts, and the second provides the target concept. The rest of the images are swapped ones ("Obj" denotes Object).

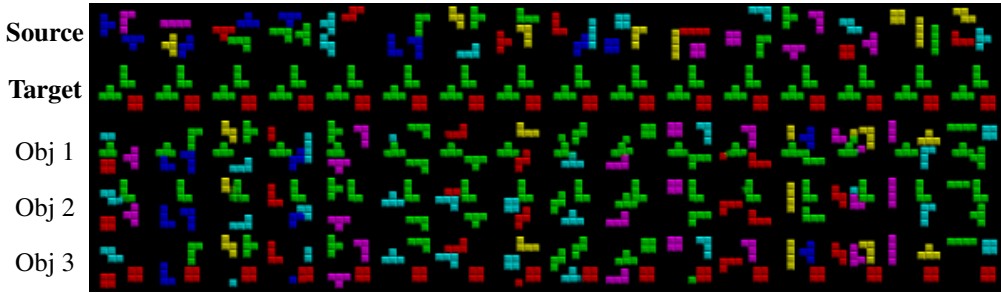

Swapping concept token on Teris

Figure 21: Visualization of swapping concepts on Teris. The images in the first row provide source concepts, and the second provides target concept. The rest of the images are swapped ones ("Obj" denotes Object).