# OpenReview forum: "Visual Concepts Tokenization"
_NeurIPS.cc/2022/Conference — NeurIPS 2022 Accept_

### Official Review · Reviewer_yDux · 2022-07-05

**Rating:** 5
**Confidence:** 3
**Soundness:** 3 good
**Presentation:** 3 good
**Contribution:** 3 good

**Summary:**

This paper proposes a new transformer-based method for learning a set of prototypes that describe different visual concepts. It leverages the cross-attention/self-attention mechanism to model the interactions between visual inputs and trainable prototypes, and presents a concept disentangling loss to encourage the learning of their correlations. Experimental results on several 3D datasets demonstrate the usefulness of the proposed method.

**Questions:**

(1) Please consider experiments with data from broader domains, and also validate the effectiveness of the method via applications such as image editing.

(2) Please justify the unique concepts learned from the proposed method, and the techniques for attributing prototypes to concepts.

(3) Please provide the details about the mechanism and impacts of the dataset-specific prototypes.


**Limitations:**

I did not find significant negative societal impacts in this work.

**Strengths And Weaknesses:**

This paper has the following strengths:
+ It is an interesting idea to leverage the attention mechanism for decomposing visual inputs into unique concepts.

+ The proposed method is able to learn visual decomposition in an unsupervised manner.

+ The paper provides extensive ablation study and analysis with different settings, and the concept disentanglement based on language is also interesting.

However, there are also weaknesses that can not be overlooked:
- The paper only carries out experiments on synthetic datasets, and the image size is typically very small (i.e., 64x64). It is unclear how well the method could generalize to broader scenarios, for example, prior studies (e.g., [20, 37]) typically consider naturalistic images with more complicated visual scenes.


- A major claim of the method is to learn disentangled prototypes that correspond to unique concepts. The paper mainly demonstrates the effectiveness of disentanglement based on information-based metrics, and it is difficult to validate if the disentangled representations are truly associated with independent visual concepts.

- Besides the targeted concept prototypes, the method also learns a set of dataset-specific prototypes Y. This is relatively counterintuitive, considering that the visual concepts are supposed to represent a diverse range of visual scenes. What do these dataset prototypes learn? And how important are they to the overall method?

- In addition to removing/adding concepts or swapping concepts between images, is it possible to utilize the proposed method for image editing? I am aware of the qualitative results shown in the supplementary materials, however, they are either limited to specific domains (synthetic data) or not very impressive (causing distortion or no significant effects on facial images).

---

> ### Author Response · Authors · 2022-08-02
> **To Reviewer yDux:**
>
> Thanks for providing constructive comments. Your concerns are addressed below.
>
> **W1 & Q1**: Thanks for your suggestion. To demonstrate that VCT can generalize to broader scenarios and validate the effectiveness in editing applications, we conduct the following experiments: (i) we apply VCT on real-world dataset KITTI, and also MSCOCO datasets **(224x224)** and (ii) We combine VCT with pretrained GAN to discovering the latent disentangled directions for editing. As mentioned in the common response (to all reviewers), disentanglement in the real world is still **quite challenging** [17]. Compared to synthesized data, the real-world data contains more diverse and unlimited scene variations, the total number of concepts is large and unknown, and the number of concepts is image specific. However, we find that **VCT still** produces some **promising** results on those two datasets. The results and implementation details are shown in Appendix B.5. We also find that VCT can combine with pretrained GAN, which can further unleash the power of VCT in the real world. It can work well and find some disentangled concepts on ImageNet (BigGAN), and LSUN cat/church (StyleGAN) with **larger image size (i.e., 256x256)**. We think those promising results can inspire the way to totally solve disentanglement in the real world.
>
> **W2 & Q2**: Generally, in disentangled representation learning literature, the metrics used in our paper (DCI disentanglement, MIG, betaVAE score, Factor VAE score) are commonly used and well accepted for evaluating the disentangled representation [5, 8,9,22,27,28,29,30,36,41]. These metrics not only evaluate how to disentangle between tokens but also how the tokens are associated with independent visual concepts [8,14,27,22]. Among those metrics, only MIG is information-based, but others are not.
> In our paper, following the previous works [8,14, 22,27], these metrics are computed by using the concept ground truth labels. The matching degree of concept tokens and independent concept ground truth labels is also evaluated by these metrics.
>
> We suppose the meaning of “justify the techniques for attributing prototypes to concepts’’ is “why the concept prototypes can learn these concepts.” If we have a misunderstanding, please give us a more detailed description. In a nutshell, to get disentangled concept tokens from a given image, 1. the process of extracting those tokens should be **independent** (there is no interference between the process of extracting concept tokens), 2. ensure that each concept token can only reflect one kind of visual concept variation. These two key points can be well implemented by using cross attention operation (for point 2) without self-attention operation (for point 1) in extracting concept tokens. Further, the proposed Concept Disentangling Loss encourages the **mutual exclusivity** between the visual variations caused by modifying different concept tokens. In the cross-attention operation, each concept prototype encodes each kind of concept variation, and each concept variation corresponds to a visual concept.
>
>
> **W3 & Q3**: The prototype Y is not so important for VCT. Even replacing the Concept Detokenizer with the original transformer (see ”Transformer DeTokenizer“ in Table 2, which does not have Y), the performance does not have catastrophic drops. Please note that the use of the prototype Y makes the Concept DeTokenizer symmetric with Concept Tokenizer since a symmetric autoencoder architecture is a common design in the literature (e.g., VAE). The prototype Y behaves as placeholders or containers to allow the concept tokens to inject information into the decoding process.
>
> **W4**: Thanks for pointing out it and inspiring us to use VCT editing images. The answer is yes. Considering these works [20,37] that utilize pretrained GANs without applying reconstruction loss, thus we propose a method for using VCT by taking pretrained GANs as a decoder and discarding the reconstruction loss, as shown in Figure 10 in the Appendix, our VCT for editing achieves promising results on more challenging datasets including ImageNet and LSUN cat and church and FFHQ.

---

> > ### Comment · Reviewer_yDux · 2022-08-06
> > **To Paper362 Authors**
> >
> > I thank the authors for proving the additional results on naturalistic images, which show the generalizability of the proposed method. Regarding my second question, I was asking if there is a way to verify the correlation between concept tokens (i.e., prototypes) and visual concepts. Currently, the concept tokens are implicitly learned, and it is unclear which concepts they encode (e.g., token 1 corresponds to green color).

---

> > > ### Author Response · Authors · 2022-08-08
> > > **To Reviewer yDux:**
> > >
> > > Thanks for approbating the generalizability of the proposed method. Verifying such correlation is a good question for evaluation in the disentangled representation literature, which is an important evaluation aspect that the disentanglement metrics already considered (MIG)[8],(DCI)[14],(FactorVAE score)[27],($\beta$-VAE score)[22] , e.g., Figure 5 in [22] and Figure 2 in [27]. Currently, flowing previous works, using the four metrics [8,14,27,22] (also used in our paper), is the common way to verify such correlation (higher metrics indicate higher correlation).

---

> > > ### Author Response · Authors · 2022-08-09
> > > **A gentle reminder**
> > >
> > > Dear Reviewer yDux,
> > >
> > > We want to send you a kindly reminder for the discussion, since the stage of discussion will be soon concluded.
> > >
> > > We thank you again for your valuable comments, and we are happy to extend our response if you have any other concerns left.
> > >
> > > Thanks.

---

### Official Review · Reviewer_zBAX · 2022-07-13

**Rating:** 7
**Confidence:** 4
**Soundness:** 3 good
**Presentation:** 3 good
**Contribution:** 3 good

**Summary:**

The paper proposes an unsupervised framework called Visual Concepts Tokenization (VCT) to extract visual concepts from concrete pixels for tackling disentangled representation learning and scene decomposition. VCT adopts a cross-attention-based tokenizer to abstract visual information into concept tokens. In addition, it also utilizes a concept disentangling loss based on the visual concept token manipulation to ensure the exclusivity of different tokens. The authors conduct extensive experiments to demonstrate the superiority of VCT against SOTA methods on multiple popular disentanglement benchmarks. Furthermore, the authors perform many ablation studies to verify the design choices of cross-attention only concept tokenizer and concept disentangling loss. Even trained without any dedicated design for scene decomposition tasks, VCT can still well decompose a scene into object-level visual representations.

**Questions:**

1. Can VCT generalize well to real-world image datasets?
2. How to decide which learned concept token corresponds to which factor of variation?

**Limitations:**

No obvious negative societal impact was found.

**Strengths And Weaknesses:**

Strengths:
1. The paper is well-written and easy to follow. The proposed VCT approach performs well on disentangled representation learning and scene decomposition tasks. It achieves SOTA performances on three datasets and beats all four types of baseline methods (VAE-based, GAN-based, pre-trained GAN-based, concept-based) by a relatively large margin.
2. As shown in the quantitative results in Figure 3, 4, and 5, VCT performs pretty well on decomposing, reconstructing, and manipulating the learned visual concepts, which demonstrates great model interpretability.
3. Even not trained with explicit object mask annotations, VCT can still abstract high-level visual representations and achieve scene decomposition by representing each object as a concept token.
4. Extensive ablation studies are conducted to confirm the design choices of image tokenizer, concept disentangling loss, concept tokenizer, and concept detokenizer. Another big advantage of VCT is that it is robust to hyper-parameter variations.

Weaknesses:
1. Except for the face dataset CeleBA, the paper conducts all experiments on synthesized datasets. CeleBA is also with limited scene variations. It is doubtful whether the proposed VCT can generalize well to real-world images such as those in ImageNet or MSCOCO.
2. From the qualitative results presented in the Appendix, for predefined $M$ visual concept tokens, only a proportion of them have explicit meanings after training. Moreover, it is unclear in the paper how to decide which learned concept token corresponds to which factor of variation.
3. The default setting used the image tokenizer and detokenizer from a pre-trained VQ-VAE. However, VQ-VAE might already learn meaningful quantized representations that correspond to visual concepts via the vector-quantization layer.
4. Section 4.3 and Section 4.4 are not well presented. The descriptions of the decomposition and recombination processes are hard to understand. Meanwhile, the caption of Figure 6 seems not to match its content, especially for Figure 6 (b). I suggest the authors further improve the presentation for scene decomposition experiments.

---

> ### Author Response · Authors · 2022-08-02
> **To Reviewer zBAX:**
>
> Thanks for providing constructive comments. Your concerns are addressed below.
>
> **W1 &Q1**: Thanks for pointing out this. To verify that VCT can generalize to the real world, we conduct experiments on KITTI and also MSCOCO datasets. As mentioned in the common response (to all reviewers), disentanglement in the real world is still **quite challenging** [17]. Compared to synthesized data, the real-world data contains more diverse and unlimited scene variations, the total number of concepts is large and unknown, and the number of concepts is image specific. However, we find that **VCT still** produces some **promising** results on those two datasets. The results and implementation details are shown in the Appendix. We also find that VCT can be combined with pretrained GAN, which can further unleash the power of VCT in the real world. It can work well and find some disentangled concepts on ImageNet (BigGAN), and LSUN cat/church (StyleGAN). The results and implementation details are shown in Appendix B.5 (highlighted in blue). We think those promising results can inspire the way to totally solve disentanglement in the real world.
>
> **W2 & Q2**: Thanks for pointing this out. We provide the details on how to decide which learned concept token corresponds to which factor of variation. The details can be divided into the following two steps:
> (i) locating the meaningful tokens: we present a method to identify these meaningful concept tokens in the Appendix. As shown in Figure 2 in the Appendix, we calculate the variance of concept tokens across a batch of instances and obtain a variance vector for each concept. Then, we calculate the l2 norm of the variance vector. The norms of meaningful concept tokens are significantly larger than the rest of the tokens.
>
> (ii) identifying the factor of variation for each meaning token. If  the ground truth concept is available,  we take a set of (at least two) images (with only one target concept different) and extract their concept tokens,  the token with the largest variance represents the target concept . If the ground truth is not available, we swap the concept token of two different images and then manually observe the change of the decoded images, and determine the concept of this token. This is quite similar to previous works in identifying the factor of variation by traversing the disentangled representation [8,9,27,28,29,30,36,41].
>
> **W3**: Thanks for this suggestion. We conduct an evaluation on the representation of pretrained VQ-VAE by regarding the quantized vector as concept tokens. As the results are shown below, the representation of pretrained VQ-VAE (Pretrained VQ-VAE) has almost no disentanglement. To further verify the effectiveness of VCT, we take a randomly initialized vanilla AE (AE + VCT) and pretrained vanilla AE (pretrained AE) as Image Tokenizer. VCT slightly drops and even has gains on MIG compared to the default setting (pretrained VQ-VAE + VCT). These results demonstrate that the power of learning visual concepts is not dependent on pretrained VQ-VAE.
> | Model| MIG |
> | :-----:| :----: |
> | pretrained VQ-VAE | 0.0185 |
> | AE + VCT | 0.484 |
> |pretrained AE + VCT | 0.560 |
> |pretrained VQ-VAE + VCT| 0.525 |
>
> **W4**:Thanks for pointing this out. We agree with you that 4.3 and 4.4 are not well presented here due to space limitations. We have also modified the corresponding part in our revised version (highlighted in blue) and added more details in the appendix (highlighted in blue). As for figure 6, we reformed it in the main paper and put the original figure 6 with the refined descriptions into the Appendix.

---

> > ### Comment · Reviewer_zBAX · 2022-08-08
> > **Thanks a lot for the detailed response and revised version**
> >
> > The authors provided a detailed response to my questions and concerns. Their response solves most of my concerns. For the concern about real-world generalization, they offered more case studies on KITTI and MSCOCO. The real-world cases on the KITTI dataset look pretty impressive to me. They also solved the presentation issue in Section 4.3 and Section 4.4, now Figure 6 is much easier to understand than before. From the additional experiment on VQ-VAE, they demonstrated that VQ-VAE itself has almost no effect on disentanglement. The only concern I have now is about the procedure of identifying the factor of variation for each meaning token. From my point of view, the whole process is more of a manual step rather than decided by an algorithm automatically. In summary, I still believe this is a pretty good paper that makes enough contribution to the field of disentangled representation learning. Therefore, after reading both the author response and other review comments, I decided to keep my original rating.

---

> > > ### Author Response · Authors · 2022-08-08
> > > **To Reviewer zBAX:**
> > >
> > > Thanks a lot for appreciating our work and the generalizability of the proposed method. In addition, thanks a lot for your acknowledgment of our rebuttal.
> > >
> > >
> > >
> > >
> > >
> > >
> > >
> > > For the remaining concern, we may not have made it clear in our response before that this process is completely automated when GT labels are available. Specifically, for each factor of variation, we sample a set of (at least two) images with only this factor of variation different, but others kept the same and extract concept tokens of these images, the token with the largest variance represents that it corresponds to this factor of variation. No matter the operation of sampling images, encoding, calculating variance, and taking the largest value, all parts of this process can be automated. Also, please note that the above identifying process does not require locating the meaningful tokens first because the variance of the meaningless token itself is almost 0. In the absence of GT, as far as we know, there is currently no method in the literature (e.g., neither all kinds of VAEs nor DisCo) that can automatically find these correspondences between representation and factors of variation. In this context, this is a very interesting and worthwhile question to explore.

---

> > > ### Author Response · Authors · 2022-08-09
> > > **A gentle reminder**
> > >
> > > Dear Reviewer zBAX,
> > >
> > > We want to send you a kindly reminder for the discussion, since the stage of discussion will be soon concluded.
> > >
> > > We thank you again for your valuable comments, and we are happy to extend our response if you have any other concerns left.
> > >
> > > Thanks.

---

> ### Author Response · Authors · 2022-08-08
> **We are looking forward to your feedback**
>
> Dear reviewer zBAX,
>
> We would appreciate your feedback, and would be happy to address any your remaining concerns.
>
> Best,
>
> Paper362 Authors

---

### Official Review · Reviewer_YRhc · 2022-07-19

**Rating:** 5
**Confidence:** 3
**Soundness:** 3 good
**Presentation:** 3 good
**Contribution:** 2 fair

**Summary:**

This paper proposes a new unsupervised transformer architecture for learning disentangled visual concepts. Main contributions include the novel transformer-based architecture to represent images as a set of tokens, each reflecting a visual concept; and a concept disentangling loss, which asks the model to predict mutated concept tokens. Experiments are conducted on Shapes3D, MPI3D, Cars3D, which demonstrate better disentangling and scene decomposition capabilities of the propose model compared with other existing works. When using CLIP encoder as image tokenizer, the framework also enables language-aligned disentanglement.

**Questions:**

1. From Table 2, how L_dis will affect the result if AE / VQ-VAE are pre-trained?
2. From Table 2, are the sensitivity results (batch sizes, token numbers) dependent on the datasets? For example, I would expect more complex scenarios are more vulnerable with respect to the number of tokens.
3. Has transformer-based architecture used previously for learning disentangled visual representation? What are the main points to highlight in VCT in comparison to them?

**Limitations:**

The key limitation is the lack of convincing results on real-world scenarios, considering the conclusion are most experimental rather theoretical. In addition, it is not always clear why such an architecture leads to better disentangled representation in general, due to the lacking in a good motivation.

**Strengths And Weaknesses:**

S1: The propose architecture seems principle, and as the paper shows can be used in combination with different model architectures.

S2: The concept disentangling loss is beneficial to further learn disentangled visual representation, without additional manual annotations.

S3: Quantitative and qualitative results are decent in demonstrating the quality of the obtained disentangled representation on the commonly used datasets.

S4: The presentation is smooth and easy to follow.

W1: Some parts of the writing may be under-supported by evidence. See Questions.

W2: Disentanglement results are mostly on simulated datasets. It is not completely clear how the proposed method would transfer to real-world datasets, such as KITTI. The results on CelebA in the supplementary is not always promising, with attributes tangled in some cases.

W3: The design may seem not very well motivated. Contributions are mostly on the architecture side, with no strong intuition why or whether such a design leads to superior performance in general.

---

> ### Author Response · Authors · 2022-08-02
> **To Reviewer YRhc (Part 2):**
>
> **W3**: Thanks for your suggestion. We have some discussion on the motivation of our design in the introduction of our paper. We should emphasize more on it. In a nutshell, to get disentangled concept tokens from a given image, we have two key points. 1. the process of extracting those tokens should be **independent** (there is no interference between the process of extracting concept tokens). 2. Ensure that each concept token can only reflect one kind of visual concept variation. These two key points can be well implemented using cross-attention operation (for point 2) without self-attention operation (for point 1) in extracting concept tokens. Further, the proposed Concept Disentangling Loss encourages the **mutual exclusivity** between the visual variations caused by modifying different concept tokens. Lastly, The extracted concept tokens should be **complete** to represent the image, i.e., the image can be well reconstructed from the concept tokens. This inspires us to adopt an auto-encoder architecture. Considering the disordered nature of concepts, the ranking order of tokens should not carry any information, so we do not adopt positional embedding for concept tokens and prototypes.

---

> > ### Comment · Reviewer_YRhc · 2022-08-08
> > **Thanks for the response**
> >
> > I have read the response and other reviews carefully.
> >
> > Two concerns remain:
> > - from the practical side, it is not always clear how the proposed disentanglement transfers to real-world cases, where the number of visual concepts can be non-trivial to define in advance. It may not always hold that different instances from the same domain would share the same number of visual tokens. However, this may not be a specific issue for this paper but not uncommon for disentangle learning works.
> > - from the theoretical perspective, there seems a lacking of analytic intuition while the main contribution is architectural. Experiment evidence is valid while not always convincing.
> >
> > I am therefore not in particular impressed but wouldn't argue if the paper gets in, considering the approach is generic and possibly principal.

---

> > > ### Author Response · Authors · 2022-08-08
> > > **To Reviewer YRhc**
> > >
> > > Thanks for your reply. Your remained concerns are addressed below:
> > >
> > > - As approbated by Reviewer yDux and Reviewer zBAX, the generalizability of our work to the real world is well demonstrated via results on the following representative and popular real-world datasets (MSCOCO, KITTI, LSUN cat/church, FFHQ, ImageNet) in Appendix B.5.
> > > We want to remind you that our statement ("the total number of concepts is large and unknown, and the number of concepts is image specific'') is to emphasize the difficulties existing in the real-world dataset, but does not mean that our method has those limitations ("the number of visual concepts can be non-trivial to define in advance'', "It may not always hold that different instances from the same domain would share the same number of visual tokens.'').
> > > Please note that we don't assume that the specific value of the number of visual concepts is known in our method. In addition, 1) Even though the number of visual concepts is smaller than GT concept numbers, our method does not catastrophic fail but still works to some extent (see results of ``tokens number = 3'' in the second Table of our response). 2) Even though the number of visual concepts is unknown, we still have promising results on real-world datasets like KITTI, which is impressive to Reviewer zBAX.
> > >
> > > - From a theoretical perspective, we provide an analytic intuition here. Independence is one of the key requirements for disentangled representation in the literature [8, 5, 22, 27, 41]. In some prior works [8, 9, 5,22, 27, 41], the Total Correlation ($p(z_1,\dots,z_m) =  \Pi_i p(z_i)$, where $[z_1,\dots,z_m]$ is the representation derived by the encoder) was regarded as a theoretical guarantee and constraint of the independence applied on extracted representation.
> > > In our paper, we constrain the independence of the extraction process: no interference between the process of extracting concept tokens, i.e., a concept token is the function of only the corresponding prototype, which means that other prototypes do not affect this concept token. We provide proof of such independence as an analytic intuition.
> > >
> > > Target of proof: the concept token $c_i$ is the function of prototype $p_i$ but is independent of other prototypes $p_j, j\neq i$.
> > >
> > >
> > > ***Proof*** We denote the output of the cross-attention operation as $U \in \mathbb{R}^{M\times D_v}$,
> > > $$
> > > U = \text{cross-attention}(P,Z,Z) = softmax(\frac{1}{\sqrt D_q}Q_PK_Z^T)V_Z, $$
> > > where $Q_P=PW_Q \in \mathbb{R}^{M\times D_q}$ is the projection of the prototypes $P \in \mathbb{R}^{M\times D}$ via projection parameters $W_Q \in \mathbb{R}^{D\times D_q}$. $p_i, i=1,2\dots,M$ is the $i$-th row of $P$.
> > > $K_Z=ZW_K \in \mathbb{R}^{N\times D_q}$ is the projection of the image tokens $Z\in \mathbb{R}^{N\times D}$ via projection parameters $W_K \in \mathbb{R}^{D\times D_q}$.
> > > $V_Z = ZW_V \in \mathbb{R}^{N\times D_v}$ is the projection of the image tokens $Z\in \mathbb{R}^{N\times D}$ via projection parameters $W_V \in \mathbb{R}^{D\times D_v}$.
> > >
> > > Since the softmax operator here applies the softmax function on every row of its input matrix, the output of the cross-attention operation can be reformulated as:
> > > $$
> > > U = softmax(\frac{1}{\sqrt D_q}PW_QK_Z^T)V_Z $$
> > > $$\quad= [softmax(\frac{1}{\sqrt D_q}p_1W_QK_Z^T)V_Z, \dots, softmax(\frac{1}{\sqrt D_q}p_MW_QK_Z^T)V_Z]
> > > $$
> > > We use $f$ to denote other operations (layer norm, feed-forward network, skip connection, which is operated on token-level) followed the cross attention operation. Therefore, we can derive the output (concept tokens $C\in\mathbb{R}^{M\times D_v}$) of the cross-attention layer:
> > > $$ [c_1, \dots,c_M]  = C = f(U) = f(softmax(\frac{1}{\sqrt D_q}Q_PK_Z^T)V_Z)$$
> > > $$\quad= [f(softmax(\frac{1}{\sqrt D_q}p_1W_QK_Z^T)V_Z), \dots, f(softmax(\frac{1}{\sqrt D_q}p_MW_QK_Z^T)V_Z)]
> > > $$
> > > Therefore, we have $c_i = f\left(softmax(\frac{1}{\sqrt D_q}p_iW_QK_Z^T)V_Z\right)$. Therefore, the changes of prototype $p_j$ will not influence $c_i$, if $i \neq j$. Similarly, since $C$ is the query in the next cross-attention layer, this also holds for the multi-layers case. Therefore, there is no interference between the process of extracting concept tokens.

---

> > > ### Author Response · Authors · 2022-08-09
> > > **A gentle reminder**
> > >
> > > Dear Reviewer YRhc,
> > >
> > > We want to send you a kindly reminder for the discussion, since the stage of discussion will be soon concluded.
> > >
> > > We thank you again for your valuable comments, and we are happy to extend our response if you have any other concerns left.
> > >
> > > Thanks.

---

> ### Author Response · Authors · 2022-08-03
> **To Reviewer YRhc (Part 1)**
>
> Thanks for providing constructive comments. Your concerns are addressed below.
>
> **W1&Q1**: Thanks for pointing out this. We find that we have a typo here. In Table 2, the term “VQVAE” should be “pretrained VQVAE.” Therefore, comparing the following two cases: (i) AE + VCT vs AE + VCT w/o $\mathcal L_{dis}$; (ii) pretrained VQVAE + VCT vs pretrained VQVAE + VCT w/o $\mathcal L_{dis}$ The results support the statement in the main paper: without $\mathcal L_{dis}$, VCT significantly drops but can still learn a disentangled representation to some extent. In addition, we also provide the results of “pretrained AE + VCT w/o $\mathcal L_{dis}$” below, which also conforms to our statements.
> | Models | MIG | DCI |
> | :-----:| :----: | :----: |
> | pretrained AE + VCT |0.560 | 0.849 |
> | pretrained AE + VCT w/o $\mathcal L_{dis}$ | 0.180 |  0.674 |
>
>
> **W1&Q2**: Yes, for a dataset of more complex scenarios (i.e., with more GT concepts), a larger token number M is needed in VCT. Specifically, M should be no smaller than the number of GT concepts. Since we apply the disentangling loss inside each batch, to ensure the diversity inside a batch, the batch size should also be no smaller than the number of GT concepts. As the number of GT concepts is usually relatively small in the synthesized data, e.g., 6 for Shape3D, the setting of M and batch size are often satisfied. In this sense, the statement “more complex scenarios are more vulnerable with respect to the number of tokens’’ is held.
> Furthermore, in order to verify this, we add an experiment on Shapes3D with token number M and batch size set to 3, which are smaller than the number of GT factors/concepts number 6. As the table shows below, the performance significantly drops. However, if the concept number is already >= GT factors/concepts number, the performance is robust to the concept tokens number (See Table 2). Therefore, this phenomenon supports the claim that “batch sizes, token numbers influence sample diversey… but VCT is still robust to batch sizes, token numbers”, under the condition that token number/ batch size>= GT factors/concepts number. We also added this condition to the revised version (highlighted in blue).
>
> | Settings| MIG | DCI |
> | :-----:| :----: | :----: |
> | tokens number = 3 | 0.450 | 0.599 |
> | tokens number = 10 | 0.533 |  0.867 |
> | tokens number = 20 | 0.525 |  0.884 |
> | tokens number = 30 | 0.493 |  0.885 |
>
> | Settings| MIG | DCI |
> | :-----:| :----: | :----: |
> | batchsize = 3     |     0.418  | 0.790 |
> | batchsize = 16     |     0.497 |  0.862  |
> | batchsize = 32     |     0.525 |  0.884  |
> | batchsize = 64     |     0.535 |  0.900 |
>
> **W1&Q3**: As claimed in our paper, to the best of our knowledge, we are the first transformer-based architecture for learning disentangled visual representation. Here we want to emphasize that, in VCT, the operation of obtaining concept tokens (retrieving from input via cross attention) is well aligned with the kernel requirement of getting disentanglement representation, i.e., the process of extracting different concepts should be independent.
>
> **W2**: Thanks for this suggestion. To make clear how our VCT would transfer to real-world datasets, we conduct experiments on KITTI and also MSCOCO datasets. As mentioned in the common response (to all reviewers), disentanglement in the real world is still **quite challenging** [17]. Compared to synthesized data, the real-world data contains more diverse and unlimited scene variations, the total number of concepts is large and unknown, and the number of concepts is image specific. However, we find that **VCT still** produces some **promising** results on those two datasets. The results and implementation details are shown in Appendix B.5 (highlighted in blue). We also find that VCT can be combined with pretrained GAN, which can further unleash the power of VCT in the real world. It can work well and find some disentangled concepts on ImageNet (BigGAN), LSUN cat/church (StyleGAN), and FFHQ (StyleGAN). We think those promising results can inspire the way to totally solve disentanglement in the real world.

---

> ### Author Response · Authors · 2022-08-08
> **We are looking forward to your feedback**
>
> Dear reviewer YRhc,
>
> We would appreciate your feedback, and would be happy to address any your remaining concerns.
>
> Best,
>
> Paper362 Authors

---

### Author Response · Authors · 2022-08-02
**To All:**

**We thank all the reviewers for the positive feedback and constructive comments. VCT is principal (Reviewer YRhc) and interesting (yDux), and achieves decent (Reviewer YRhc) and SOTA (Reviewer zBAX) performance.**

**The main common concern is whether VCT can be generalized to real-world data or not.** Please note that disentanglement in the real world is still **quite challenging** [17]. Compared to synthesized data, the real-world data contains more diverse and unlimited scene variations, and the total number of concepts is large and unknown. The number of concepts in a single image is image specific. The previous SOTA method can only address CeleBA [8,9,27,28,29,30,41] (with limited scene variations as pointed out by Reviewer zBAX). Please note that [20,37] do not target learning disentangled representation but discovering the latent direction of the pretrained GAN instead. Besides, they highly rely on pretrained GANs.\
We generalize VCT to more complex real-world datasets MSCOCO and KITTI, and find that **VCT still** produces some **promising** results on those two datasets. The results and implementation details are shown in Appendix B.5 (highlighted in blue). Furthermore, inspired by the suggestion of Reviewer yDux and DisCo [36], we combine the tokenizer of VCT with a pretrained GAN, resulting in a new architecture. As the results are shown in Appendix B.5 (highlighted with blue), this new architecture can learn a lot of disentangled **concepts on ImageNet (BigGAN), LSUN cat/church (StyleGAN), and FFHQ (StyleGAN).** \
Those observations all show the potential capability of VCT in real-world scenarios. However, we do not mean this challenging problem is totally solved by VCT, as some failure cases are shown in Appendix B.5 (highlighted in blue). We think that VCT, with a novel architecture, is promising in addressing this challenge, which can be beneficial to this research community.

---

### Meta-Review · Area_Chair_667H · 2022-08-27

**Recommendation:** Accept
**Confidence:** Less certain

**Metareview:**

This paper proposes an unsupervised transformer based framework called Visual Concepts Tokenization (VCT) to extract visual concepts from concrete pixels for tackling disentangled representation learning and scene decomposition. Experiments on several popular datasets validated the effectiveness of VCT on the tasks of disentangled representation learning and scene decomposition in which VCT outperforms the previous works significantly. Reviewers generally agree the proposed VCT framework is novel and the empirical results are promising (though the results on real-world images seem not as strong as the synthesized data and clearly there is a room to improve on the real-world image data). Authors did a great rebuttal job in making extensive efforts to make revision and give comprehensive answers in response to the reviewers's concerns. Overall, this is a solid paper that has enough contributions to the disentangled representation learning topic and thus is recommended to accept.

**Award:**

No

---

### Decision · Program_Chairs · 2022-09-14

Accept